# Estimation of Lassa fever incidence rates in West Africa: Development of a modeling framework to inform vaccine trial design

Sean M. Moore[1]*, Erica Rapheal[2], Sandra Mendoza Guerrero[3¤], Natalie E. Dean[4], Steven T. Stoddard[3¤]

1 Department of Biological Sciences, University of Notre Dame, Notre Dame, Indiana, United States of America, 2 Independent Consultant, Minneapolis, Minnesota, United States of America, 3 Emergent BioSolutions, Inc., Gaithersburg, Maryland, United States of America, 4 Department of Biostatistics & Bioinformatics, Emory Rollins School of Public Health, Atlanta, Georgia, United States of America

¤ Current address: Bavarian Nordic, Inc., Durham, North Carolina, United States of America
* smoore15@nd.edu

## Abstract

### Background

Lassa fever (LF) is an acute viral hemorrhagic disease endemic to West Africa that has been declared a priority disease by the World Health Organization due to its severity and the lack of a vaccine or effective treatment options. Several candidate vaccines are currently in development and are expected to be ready for phase III field efficacy trials soon. However, most LF cases and deaths are believed to go unreported, and as a result we lack a clear understanding of several aspects of LF epidemiology and immunology that are critical to the design of vaccine efficacy trials.

### Methods

To help guide vaccine trial design and trial site selection we estimated the force of infection (FOI) from rodent hosts to humans in all 1st and 2nd administrative units in West Africa from published seroprevalence studies. We next estimated LF reporting probabilities using these FOI estimates and LF case and death reports and then projected FOI in all admin1 and admin2 areas without seroprevalence data. We then extrapolated age-specific LF incidence rates from FOI estimates under different assumptions regarding the level of protection against reinfection among seropositive and seronegative individuals with a history of prior infection.

### Results

Projected FOI estimates and modeled annual LF incidence rates indicate that Sierra Leone, southern Guinea, and a few areas within Nigeria would likely experience the highest LF case incidence rates for a vaccine trial. Estimated LF incidence rates were

**Data availability statement:** All of the data used in this study, along with model code to reproduce our results are available at https://github.com/mooresea/lassa-model.

**Funding:** This research was supported, in part, by a contract to Emergent Biosolutions from the Coalition for Epidemic Preparedness Innovations (CEPI) and by a National Institutes of Allergy and Infectious Diseases/National Institutes of Health (NIAID/ NIH) contract, HHSN272201700077C. The funders played no role in the design, data collection, analysis, decision to publish, or preparation of the manuscript.

**Competing interests:** I have read the journal's policy and the authors of this manuscript have the following competing interests: SMM, ER, and NED report consulting fees from Emergent Biosolutions.

highly sensitive to assumptions about Lassa immunology, particularly the frequency of seroreversion among previously infected individuals and the extent to which seroreverted individuals retain protection against reinfection and more severe disease outcomes.

### Conclusions

Our spatial LF incidence rate estimates, along with the interannual and seasonal variability in these estimates and estimates of baseline seroprevalence, could be used for vaccine trial site selection, choosing the target population (e.g., age and serostatus), and maximizing a trial's statistical power.

## Author summary

Lassa fever virus infects an estimated 0.9 to 4.3 million people and kills thousands annually in West Africa. Incidence rates appear to be highly spatially heterogeneous within the endemic region; however, the true nature is uncertain due to significant surveillance gaps. We modeled Lassa Fever disease incidence at a sub-national scale throughout West Africa to inform the design of vaccine efficacy trials. We find considerable spatial heterogeneity in incidence rates, with the highest rates concentrated in Sierra Leone, Guinea, and a few areas of Nigeria. Even though we estimate that <1% of infections are reported, our estimates also indicate that using symptomatic LF as a primary endpoint will require tens of thousands of trial participants to demonstrate vaccine efficacy. Our work highlights data gaps and uncertainties related to the ecology and epidemiology of LASV that limit our ability to estimate and predict disease incidence.

## Introduction

Lassa fever (LF) is an acute viral hemorrhagic illness endemic to West Africa. LF is caused by infection with Lassa virus (LASV), an arenavirus that circulates in rodent populations but can spill over to human populations [1]. Transmission is thought to occur through human contact with rodents either due to infestation of human residences or processing for food [2]. Importantly, direct human-to-human transmission in nosocomial settings also occurs, creating a potential for wider spread of the virus to naive populations [2]. Most LASV infections are presumed to be asymptomatic or sub-clinical, but up to 20% of infections are thought to result in symptoms ranging from mild (i.e., fever, headache) through to severe (i.e., coma, acute renal failure) or fatal hemorrhagic illness [2]. Case fatality rates (CFRs) among reported or hospitalized LF cases are dependent on the likelihood of case detection, but can reach 60% when detection of severe cases is delayed. [1] In both 2022 and 2023, Nigeria reported a CFR of 18%, which indicates that the CFR remains high even after improvements to the disease surveillance and response systems [3]. Previous

studies have estimated that hundreds of thousands—and potentially as many as 0.9 to 4.3 million—LASV infections and 5,000–18,000 LF deaths occur annually in sub-Saharan Africa [4–6]. However, these estimates are highly uncertain as they were either extrapolated from limited serological studies conducted decades ago or generated from modeled spillover rates from the rodent reservoir to humans.

Despite its severity and burden, there are currently few therapeutic options for LF and no licensed vaccines [1]. In 2018, the World Health Organization (WHO) declared LF a 'Priority Disease,' and the Coalition for Epidemic Preparedness Innovations (CEPI) subsequently invested in the development of six LF vaccine candidates, four of which have entered clinical trials [7]. Individuals previously infected with LASV maintain LASV-specific CD4 + memory T cells for years after the infection has cleared [8]. This, along with animal vaccination models, suggests that vaccination will provide meaningful protective immunity against severe LF [9]. However, there are currently no immunological correlates of protection against LF and the risk of severe disease rules out the option of controlled human infection studies to evaluate the clinical benefit of a candidate vaccine [10]. Instead, field efficacy trials are necessary to determine the efficacy of vaccine candidates.

Assuming a primary endpoint of PCR-positive symptomatic LF disease, CEPI has targeted a minimum annual LF incidence rate of 1% to ensure an adequately powered trial. An ideal study site will have a minimum baseline seroprevalence level and/or evidence of ongoing circulation of the virus in the local rodent population as indicators of the presence of frequent LASV spillover. However, the baseline seroprevalence level must be low enough that existing immunity in the population does not substantially reduce the LF incidence rate. Identifying populations in which these criteria are likely to be met is the most critical consideration in Lassa vaccine trial design [11,12]. However, our understanding of Lassa epidemiology is limited by a lack of good prospective epidemiological data and inconsistent disease surveillance across West Africa. Since 2018, reported LF case counts have increased dramatically, although this is partly due to changes in surveillance and diagnostics [13]. Other than a couple of prospective community studies in Sierra Leone and Mali that tested individuals for evidence of a recent LASV infection [4,14], and a few hospital-based surveillance studies of severe LF cases [15,16], most epidemiological data on Lassa come from national surveillance programs. A majority of LF cases are not detected by current clinical surveillance systems, at least in part because common LF symptoms—such as fever, malaise, headache, and muscle pain—closely resemble those of other febrile illnesses endemic to the region [1,2,17]. Access to healthcare facilities varies considerably [18], which—along with other factors—influences the probability that symptomatic individuals will seek treatment [19,20]. In addition, the extent of national LF surveillance varies considerably from country to country in West Africa, complicating comparisons of incidence rates between different LF-endemic areas. For example, Nigeria has expanded LF surveillance over the past decade [13,21], while changes in healthcare-seeking behaviors in areas of Sierra Leone heavily impacted by the 2013–2016 Ebola virus epidemic have led to declines in the detection of febrile illnesses, including LF [16,22].

Because of the limited availability of epidemiological data for Lassa, in 2020 CEPI initiated a long-term, multi-country, prospective epidemiological study of Lassa disease and infection in West Africa [9]. This study, called Enable, is tracking over 20,000 participants across five West African countries with the goal of identifying baseline seroprevalence, LASV infection rates, LF incidence rates, serological dynamics, and individual and community-level risk factors for infection and disease. Due to the impact of the COVID-19 pandemic, data from the Enable study have not yet been published. However, preliminary findings from the study presented in an interim report suggest that LASV infection is common in several of the study locations and that prior infection does not confer lifelong immunity [23]. Epidemiological data such as those being collected in Enable are essential for understanding the current status of Lassa in the region. In the absence of detailed epidemiological data across the entire geographic range of LF, statistical models can be used to anticipate the future disease incidence for the purpose of vaccine trial planning. Such a modeling framework, based on existing epidemiological data and accounting for the factors that drive pathogen transmission and infection risk, can incorporate new epidemiological data from Enable and other studies as it becomes available. This framework can include ecological factors that influence the spatial and temporal patterns of disease and immunological factors related to immunity.

LASV is thought to infect humans through exposure to the urine or feces of the multimammate rat, *Mastomys natalensis,* which serves as the primary reservoir host [24]. As a result, LASV spillover typically occurs seasonally in rural and peri-urban areas of West Africa where agricultural practices, socioeconomic factors, and the built environment promote rodent-human interactions [25–27]. Field studies have identified several risk factors associated with *M. natalensis* abundance in villages and individual houses, but have also found that *M. natalensis* abundance and LASV seropositivity in rodents are spatially heterogeneous at multiple spatial scales, from the village- to the subnational-level [28–30]. Moreover, incidence of disease can be markedly seasonal, suggesting important variation in either rodent infection rates or contact with humans as a result of rodent population dynamics and movement patterns associated with rainfall and agricultural practices that affect food availability [25,31]. Overall, the substantial spatial and temporal variation in LASV prevalence is an important consideration for vaccine trial site selection, as potential trial locations might have different LF incidence rates despite their proximity and similar environmental conditions.

Because of the scarcity of longitudinal data, it is difficult to assess LF incidence, the duration of immunity following infection, or reinfection rates. Two longitudinal serology studies have found evidence that 3–6% of seropositive individuals seroreverted from IgG+ to IgG- between sampling periods, suggesting that infection-induced immunity may not be permanent [4,32]. Preliminary data from CEPI's Enable study also indicate that seroreversion occurs relatively frequently [9]. A further complication for our understanding of LASV infection rates is that the frequency of severe LF following infection is uncertain. It is typically stated that 80% of Lassa fever infections are asymptomatic, but data supporting this assumption is extremely limited [2].

Reliable estimates of sub-national LF incidence rates and baseline population-level immunity are needed to inform trial site selection and enable successful LF vaccine efficacy trials. In lieu of detailed geographical measurements, these epidemiological indicators can be modeled from available incidence and seroprevalence data. Here, we use an epidemiological model to estimate LASV spillover rates and the annual number of community-level LF cases at the 1st and 2nd administrative levels across West Africa. We provide sub-national estimates of baseline seroprevalence and expected age-specific LF incidence rates to help guide trial selection. We also explore the sensitivity of our estimates to different assumptions of LF immunology, including the proportion of infections that are symptomatic, the level of protection provided by prior infection, and the duration of this protection.

## Methods

### Literature review

We first conducted a non-systematic review of the LF literature to characterize key parameters relevant for vaccine trial design. A selection of studies was identified using PubMed and Google Scholar search terms including "Lassa epidemiology", "Lassa serology", etc., as well as by reviewing bibliographies of meta-analyses and review articles. Data from studies of nosocomial outbreaks were excluded, except for data on symptomatic rates. Articles were reviewed for relevant information related to LASV serology, rodent epidemiology, risk factors associated with spillover and the seasonality of transmission, and incidence and symptomatic rates. The list of articles selected from the literature review is included in S1 Table.

### Model development

We integrated publicly available epidemiological data into a unified modeling framework for LF in West Africa. We applied this model to investigate key questions regarding vaccine trial design including the expected incidence rates for trial endpoints of disease and infection, the target population (e.g., geographical location, age range, and serostatus of participants), and sample size considerations (number of sites, number of enrollees). The model was adapted from an existing model originally designed to predict spillover of LASV and model reactive vaccination strategies during an LF outbreak

[33]. To better understand the magnitude and spatiotemporal distribution of LASV spillover rates and LF incidence in endemic areas, we refined the model to focus on estimating the annual force of infection (FOI), the rate at which susceptible individuals in a population are infected. The updated model also incorporates the potential for seroreversion (seropositive individuals becoming seronegative over time due to antibody waning). FOI estimates at the 1st and 2nd administrative levels were used to estimate seasonal and interannual LF incidence rates across the study region. Rates were investigated in different potential target populations defined by serostatus and age.

In our epidemiological model, we included 14 West African countries (Senegal, Gambia, Guinea-Bissau, Guinea, Sierra Leone, Liberia, Côte D'Ivoire, Ghana, Benin, Togo, Nigeria, Niger, Burkina Faso, and Mali), plus the administrative districts in Cameroon bordering Nigeria, which encompasses the known range of LASV (Fig 1). Given the extensive

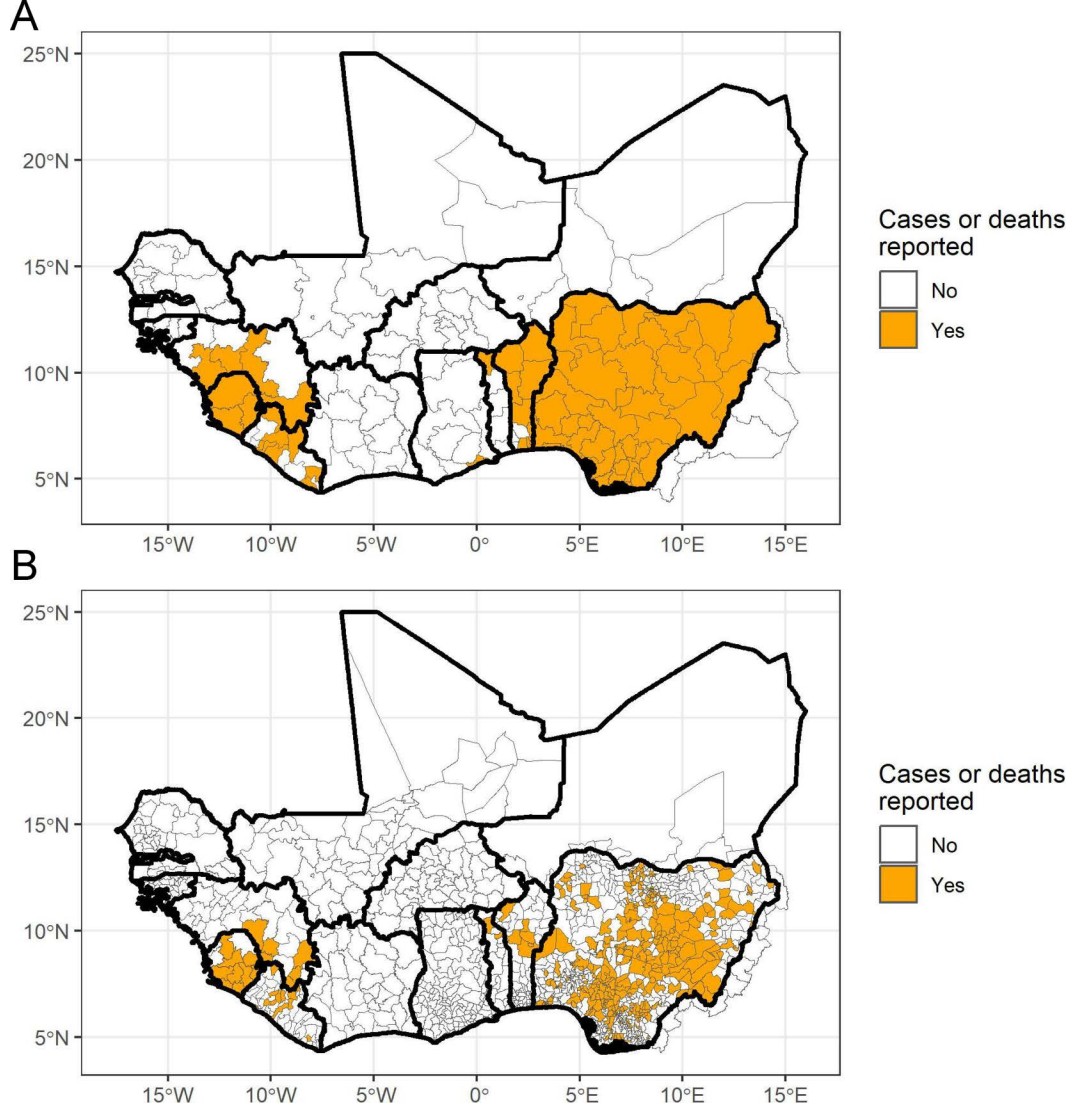

**Fig 1. Study region covering the hypothesized zone of Lassa fever (LF) endemicity.** Areas in orange are (A) 1st administrative level units and (B) 2nd administrative level units that have reported LF cases or deaths from 2010-2023. In some locations, data were only reported at the 1st administrative level. The base map layer was generated using GADM 3.6 data files which can be accessed from https://gadm.org/download_world36.html.

spatial heterogeneity in LF incidence, we collated epidemiological data at the 1st and 2nd administrative levels (admin1 and admin2) in each country within the study region. For example, in Nigeria admin1 is the state level and admin2 is the local government area (LGA) level; in Sierra Leone admin1 corresponds to the province level and admin2 is the district level. The epidemiological data included in the study comprised two types: (a) age-stratified serology data to detect evidence of past infection and (b) reports of suspected and confirmed LF cases and deaths in humans.

**a. *Serology data*.** Infection data from serological surveys was initially collated through the end of 2020 from multiple sources (including WHO outbreak reports, ProMED reports, country-level reports, and a literature search) and used in a previous analysis (see supplementary table #1 in Lerch et al. 2022) [33]. For the current analysis we searched the same sources for additional datasets through early 2023. We excluded seroprevalence studies from before 1980 or where the location of the study population could not be identified at a sub-national level. Seroprevalence studies prior to 1980 were excluded because there have been large shifts in both demographics and land usage within the study region the likely impact viral circulation within rodent populations and spillover to humans in the past 40 years. We also excluded seroprevalence studies that focused only a specific target population (generally healthcare workers) that may not be representative of the overall population in the study area due to unequal exposure to spillover or human-to-human transmission. All included studies were aggregated to the admin1 and admin2 levels.

**b. *Case and death data*.** Case and death data were obtained via the same literature search used to identify serological data. Where possible, case and death data were categorized into cases of documented or suspected human-to-human transmission, documented or suspected spillover cases, and cases of unknown origin. Cases of documented or suspected human-to-human transmission were excluded from the estimation of spillover rates. Only cases and deaths from 2010-2023 were included in our analysis, because the case/death data were used to estimate the fraction of LF cases that are reported and LF surveillance systems have changed substantially in the past decade [13,16].

**c. *Covariate data*.** To identify population-level covariates associated with LF occurrence, we used spatial datasets of environmental, climate, and socioeconomic variables that have been hypothesized to be associated with LF occurrence or transmission [13,15,34,35]. These variables included elevation, longitude, travel time to the nearest urban center [36], the Healthcare Access and Quality Index (HAQ) based on mortality from causes amenable to personal health care [37], proportion of land cover that was a tropical ecotype [38], proportion of agricultural land [39], average forest loss over the past 20 years [40], an improved housing measure [41], a poverty index (percentage of households with an International Wealth Index value below 35) [42], the occurrence of hunting for bushmeat [43], the probability of *Mastomys* occurrence [5], and the probability of LASV occurrence in *Mastomys* (S2 Table) [5]. We also considered including latitude, monthly precipitation, monthly average temperature, or monthly normalized difference vegetation index (NDVI) as explanatory variables, but each of these was highly correlated with at least one of the other covariates and therefore was not included in our analysis. Each covariate was averaged to the admin1 and admin2 level. Within the study region, the improved housing measure or *Mastomys* occurrence data was missing for some administrative units in northern Senegal, Mali, and Niger, where there is no evidence of LF occurrence, so these administrative units were excluded from our analysis. Yearly, age-specific country-level population data from 1960-2015 were obtained from UN World Population Prospects estimates and downscaled to the admin1 and admin2 levels using population raster data from Worldpop [44,45].

**d. *Model*.** A multistep process was used to model LF attack rates from recent epidemiological data: (1) estimation of the recent force of infection (FOI) in administrative units with available seroprevalence data, (2) estimation of the proportion of LF cases and deaths that were detected and reported in administrative units using both seroprevalence data and case/death data, (3) estimation of the annual LASV spillover infection rate in all administrative units with case/death data based on the underreporting estimates from the previous step, (4) projection of the annual FOI for these administrative units based on these spillover rates, (5) calculation of the population-level infection history in each administrative unit based on these FOI estimates, (6) calculation of age-specific infection attack rates and LF incidence rates in each administrative unit under several different scenarios regarding the rates of seroreversion and

the susceptibility to infection and disease among seropositive and seroreverted individuals (Table 1 and Fig 2). This modeling process was conducted at both the 1st and 2nd administrative levels. In addition, we used statistical and machine learning methods to estimate annual FOI in each administrative unit based on the covariates in S2 Table and compared these estimates to the model projections from step 4. This analysis was conducted to determine whether environmental variables associated with LF occurrence could be used to estimate LASV spillover in the absence of human serology or case data. A brief description of each step in this process is provided below, with additional details of the complete process provided in the supplementary materials (S1 Text). Although there is likely to be extensive spatial heterogeneity in LF incidence within an admin2 area (down to the village or sub-village level), case and death reports from national surveillance systems are typically aggregated to the admin1 or admin2 level. In the absence of finer scale case data, we were not able to estimate LF incidence at fine spatial scales using geostatistical modeling methods. There were also too few serological studies to reliably interpolate LASV spillover rates to finer spatial scales.

1. Estimating *the* Force *of* Infection *from* Serology Data.

Estimates of the annual FOI were obtained for each 1st or 2nd level administrative unit where serological data were available from 1980 - 2023 using a catalytic model and assuming a constant FOI ($\lambda$) over time. We estimated FOI assuming either a 0%, 3% (observed in Mali by Safronetz et al. 2017) [14], or 6% seroreversion rate (observed in Sierra Leone by McCormick et al. 1987) [4]. In the absence of seroreversion, and assuming all infected individuals develop detectable antibody levels following infection, the proportion of *a* population that will be seropositive at age *a* is determined by the FOI ($\lambda$):

$$p(a) = 1 - e^{-\lambda a}$$

(1)

If antibodies wane over time and some individuals serorevert from IgG+ to IgG-, then the proportion of the population seropositive at age a can be estimated using a reverse catalytic model:

$$p(a) = \frac{\lambda}{\lambda + \pi} \left( 1 - e^{-(\lambda + \pi)a} \right),$$

(2)

where $\pi$ is the annual seroreversion rate. Eq 2 simplifies to Eq 1 when $\pi = 0$.

Serology data for FOI estimation was obtained for 24 1st-level administrative units and 53 2nd-level administrative units. Additional details regarding how FOI was estimated from serology data are provided in S1 Text.

2. Estimating Country-specific Reporting Fractions.

For each administrative unit where the FOI was estimated from serology data in the previous step, we estimated the fraction of infections that went unreported from 2010-2023 based on the discrepancy between reported LF cases and deaths and the annual number of infections predicted by the FOI estimates from those sites under the three different seroreversion scenarios. We first estimated the location-specific fraction of infections that went underreported, the fraction that resulted in a reported LF case, and the fraction that resulted in a reported LF death using a method adapted from Perkins

**Table 1. Immunological parameters included in model sensitivity analysis, and the low, medium, and high values considered for each parameter. Lassa fever relative risk values are in comparison to fully susceptible individuals with no history of LASV infection.**

| Parameter | Low | Medium | High |
|---|---|---|---|
| Seroreversion rate | 0%/ yr | 3%/ yr | 6%/ yr |
| Lassa fever relative risk (seropositive individuals) | 0 | 0.36 | 0.53 |
| Lassa fever relative risk (seroreverted individuals) | 0.53 | – | 1 |

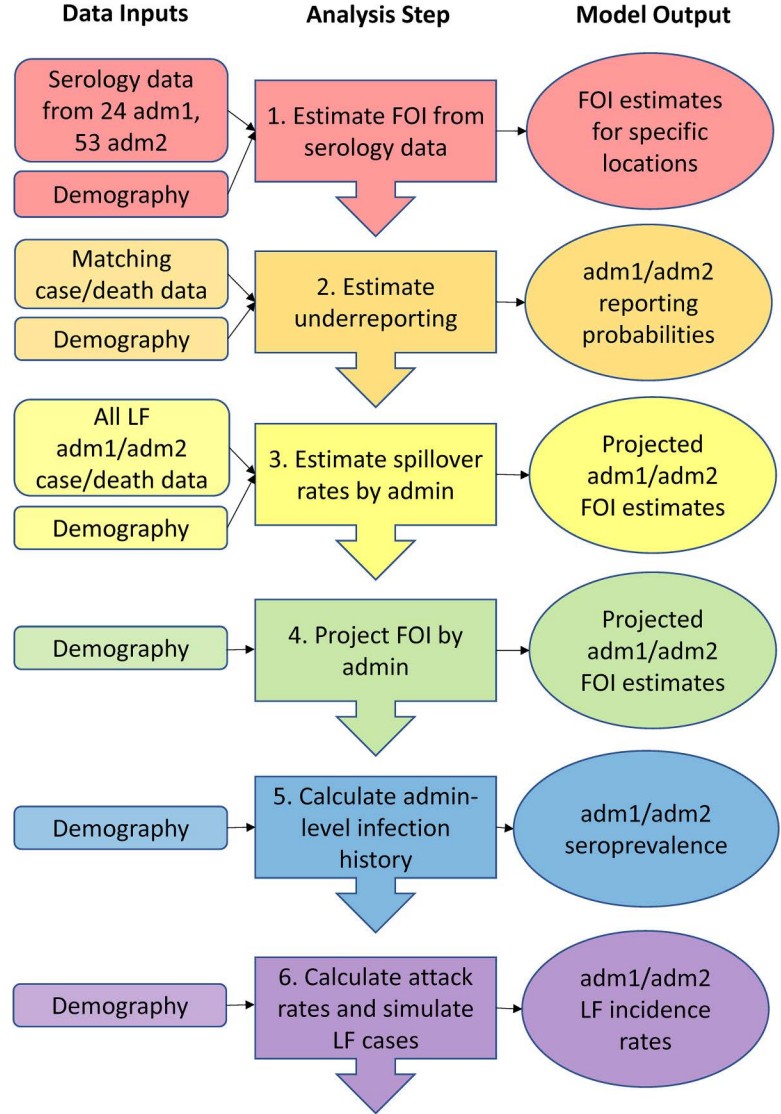

**Fig 2. Modeling framework schematic.** Our modeling framework involved seven sequential steps that result in a set of ensemble models of the FOI and annual incidence of LF in each of the 1st and 2nd administrative levels across West Africa. FOI estimates were projected and estimated for three different seroreversion rates, and LF incidence rates were estimated for 18 scenarios: 3 different seroreversion rates, 3 different assumptions about the susceptibility of seropositive individuals to disease, and 2 different assumptions about the susceptibility of seroreverted individuals to disease.

et al. (2021) [46]. The estimation process was repeated using 1000 draws from the posterior FOI estimates from step 1 to generate posterior distributions for the reporting fractions. Country-specific reporting fractions were then estimated from all available admin1 or admin2 level estimates within each country and used to extrapolate infections from LF case and death data in locations without serology data in step 3.

3. Estimating LASV spillover rates.

For each administrative unit, we next estimated the total number of annual infections, $I_{ij}$, based on the reported LF cases and deaths from 2010-2023 along with the estimated reporting fractions from the previous step using maximum likelihood estimation.

4. Projecting *the* FOI *from* estimated LASV spillover rates.

For each administrative unit where LASV spillover infections were estimated from LF case/death data in step 3, we then projected the underlying FOI that would correspond to the estimated infection rate. The projected $FOI_i$ for each administrative unit $i$ was obtained by minimizing the difference between the number of infections, $I_i$, estimated in the previous step and the expected number of infections arising from a given FOI in the reverse catalytic model from Eq 2 using the *optim* function in R. This resulted in a posterior distribution of $FOI_i$ for each admin1 and admin2 unit.

5. Estimation *of* Population-level Infection Histories.

The FOI projections generated from serology and case data in step 4 were then used to simulate population-level infection histories for each admin1 and admin2 unit. For the FOI projections, we drew 1,000 samples for each administrative unit from the posterior distribution and computed the proportion of the population that had been infected by age $a$ using the catalytic model in Eq 1, and the proportion of the population seropositive at age $a$ using the reverse catalytic model in Eq 2 for the three different seroreversion rates.

6. Estimating LASV Infection and LF Attack Rates.

The inferred population-level infection histories and FOI estimates were then used to compute the expected number of infections in each admin1 or admin2 administrative unit. We examined several different scenarios regarding the risk of seropositive or seroreverted individuals becoming reinfected and developing LF (Table 1). Reinfection of seropositive individuals, as defined by a fourfold increase in antibody titers, was observed in Sierra Leone by McCormick et al. (1987) and in the preliminary results from the ongoing Enable study [4,9]. In addition, the Enable study reported LF cases among individuals who were seropositive at baseline, indicating that prior infection does not entirely protect an individual from developing disease if they are reinfected [9]. Therefore, we considered three scenarios for the susceptibility of seropositive individuals to symptomatic infection: (a) no risk, (b) a reduced risk informed by the rates of infection observed in seronegative vs. seropositive individuals observed by McCormick et al. (1987) and Enable (relative risk (RR) = 0.53), or (c) a reduced risk informed by the relative rates of LF cases observed in individuals who were seropositive vs. seronegative at baseline in the Enable study (RR = 0.36).

Although simple reverse catalytic models generally assume that seroreverted individuals are susceptible to reinfection, an individual may still have protection against developing moderate or severe disease even if their antibody titers have dropped below the detectable limit. At present this possibility has not been addressed for LASV, so we considered two scenarios: (a) seroreverted individuals are completely susceptible to reinfection and illness, and (b) seroreverted individuals can be reinfected but have a reduced probability of developing LF based on the reduced rate of reinfection experienced by seronegative vs. seropositive individuals (RR = 0.53).

Including our three seroreversion rate scenarios, we therefore consider a total of 18 (3x3x2) scenarios regarding the role of immunity in modulating susceptibility and influencing LF attack rates. The expected annual number of infections in administrative unit $i$ were calculated from the $FOI_i$ using the reverse catalytic model in Eq 2 for each of the 18 different scenarios at both the admin1 and admin2 levels. The number of infections was multiplied by the symptomatic probability (20%) to obtain an estimate of the expected annual number of LF cases in each administrative unit.

To account for the observed seasonality in human LF cases, we fit a beta distribution to the timing of reported LF cases in Nigeria, Liberia, and Sierra Leone, and simulated the timing of LF cases as a random draw from that distribution. For countries where we could not estimate seasonality, we assumed an average of the observed seasonality in Nigeria, Liberia, and Sierra Leone.

## Modeling the FOI from covariate data

There is a large degree of uncertainty in the estimated spillover rates for administrative units that have reported only a small number of LF cases due to the large proportion of asymptomatic infections and low reporting probabilities.

Therefore, we used several statistical models to explore the relationships between our FOI estimates from step 4 and several key spatial covariates (S2 Table). These statistical regression models were fit to the projected $FOI_i$ estimates from administrative units with either serology data or reported LF case/death data (N = 77 of 164 admin1s, N = 372 of 1,375 admin2s). The fitted models were then used to predict FOI in the administrative units with no serology or case data.

Given that we have limited historical data and high uncertainty in our projected FOI estimates, we considered eight different statistical models, as well as a null model with a single FOI estimated across all administrative units. For each seroreversion scenario, we generated an ensemble model projection of FOI in each admin1 or admin2 from the eight statistical models. Ensemble weights for each of the eight models were calculated based on the performance of the individual model at predicting data withheld from the model fitting step using a ten-fold cross-validation technique. Further details on the individual statistical models and the ensemble approach are presented in S1 Text.

## Results

### Literature review

Thirty-one papers were selected for in-depth literature review. Topics of interest were Lassa serology (n = 12), rodent epidemiology (n = 10), environmental risk factors and seasonality (n = 5), and LF incidence/symptomatic rates (n = 4). Studies took place in Sierra Leone (n = 7), Guinea (n = 7), Nigeria (n = 10), Ghana (n = 1), and Mali (n = 3). Three studies included all LF cases in Africa or globally (including imported cases). Most studies were published in the last 10 years (n = 21). Seven studies were published between 2000 and 2013, and three studies were from before 2000. LASV IgG seroprevalence varied from 4% to 60%; in general, rates were higher in forest and savannah regions and lower near the coast and in the highlands [47–49,50]. Seroprevalence in rodents is similarly variable, with IgG positivity between 6–52% [28,51–54], and PCR positivity between 1–87% [55,56]. One study in Guinea found that individual villages showed some interannual variation in rodent seropositivity, but that all villages studied maintained at least 20% positivity from year to year [29]. However, LASV also appears to circulate at lower levels in other endemic areas, such as Bo, Sierra Leone, where 2.8% of rodents tested positive over a two-year period from 2014-2016 [55]. Studies of seroreversion rates in Sierra Leone and Mali found 6 and 3% of seropositive individuals, respectively, reverted to seronegative in a given year [4,28]. It is typically assumed that 80% of Lassa fever infections are asymptomatic, but data supporting this assumption is extremely limited. All studies included are shown in S1 Table.

### FOI estimates from serology

The FOI was estimated from serology data available from 1980-2023 for 24 1st-level administrative units and 53 2nd-level administrative units. FOI was estimated for at least one administrative unit in Côte d'Ivoire, Ghana, Guinea, Liberia, Mali, Nigeria, and Sierra Leone (Fig 3). Under the assumption of no seroreversion, the highest FOI at the admin1 level was in Ondo State, Nigeria (0.036/yr; 95% Credible Interval (CrI): 0.027-0.047) and the highest FOI at the admin2 level was in Moyamba District, Sierra Leone (0.063/yr; 95% CrI: 0.042-0.090) followed by Ose LGA in Ondo State, Nigeria (0.052/yr; 95% CrI: 0.037-0.071). The FOI estimates assuming annual seroreversion rates of 3% or 6% were higher than FOI estimates without seroreversion. Ondo State, Nigeria remained the highest FOI at the admin1 level, with the estimate increasing to 0.065 (95% CrI: 0.047-0.086) with a 3% seroreversion rate and 0.099/yr (95% CrI: 0.071-0.133) with a 6% seroreversion rate. At the admin2 level Ose LGA, Nigeria had the highest FOI estimate with 3% or 6% seroreversion rates, followed by Moyamba District, Sierra Leone and Esan West LGA in Edo State, Nigeria.

### Estimates of underreporting

Estimates of the probability that a LASV infection would be reported as an LF case were estimated for each of the countries with serology and LF case data (Côte d'Ivoire, Ghana, Guinea, Liberia, Mali, Nigeria, and Sierra Leone). In addition,

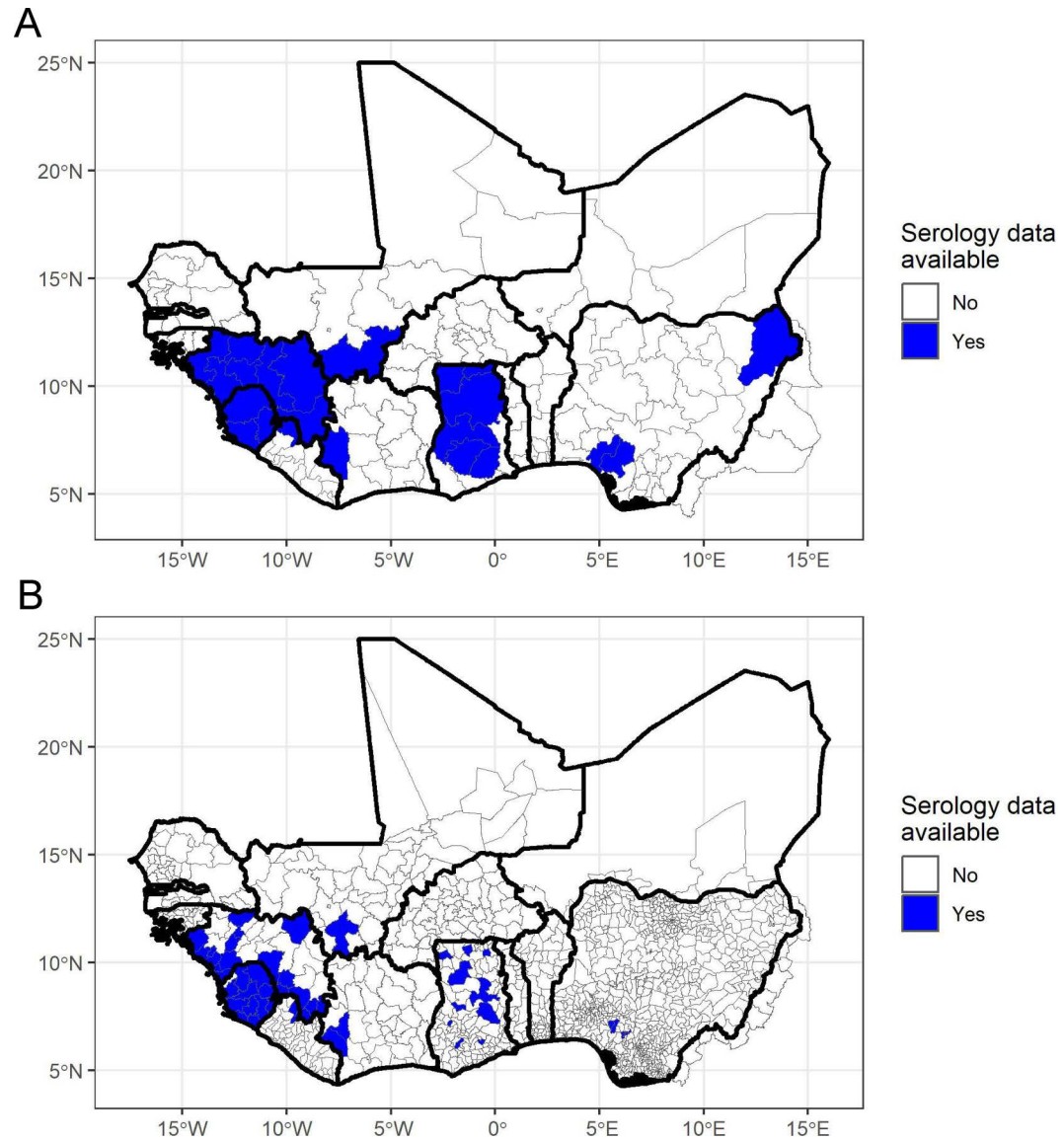

**Fig 3. Map of areas with Lassa fever (LF) serology data.** Blue areas are (A) 1st administrative level units and (B) 2nd administrative level units where age-specific serology data was available. The base map layer was generated using GADM 3.6 data files which can be accessed from https://gadm.org/download_world36.html.

we estimated the mean reporting probability across the study region, which was used to estimate LASV infections and FOI in countries that did not have serology data. Due to the higher FOI estimates with seroreversion, the probability of a LASV infection being reported was highest when we assumed no seroreversion and lowest with a seroreversion rate of 6%. Assuming a 6% seroreversion rate, the median probability that a LASV infection would be reported as a LF case or death at the admin1 level was 0.17%. The country-specific reporting probability ranged from a low of 0.20% in Ghana and Guinea, to a high of 0.88% in Nigeria. At the admin2 level, the median probability that a LASV infection would be reported as a LF case or death was 2.1%. This higher reporting probability was largely driven by the results of one serology study that found low seroprevalence in two LGAs in Edo State, Nigeria that are considered transmission hotspots [57].

## FOI projections from LF case data and reporting probabilities

The projected FOI estimates from LF case data and estimated country-specific reporting probabilities varied considerably across the study region, with evidence of spatial heterogeneity within and between countries (Figs 4, S1 and S2). In particular, there was substantial heterogeneity in FOI estimates at the admin1 and admin2 levels within Nigeria. Because serology data were only available for 3 out of 774 LGAs within Nigeria (admin2), these FOI estimates are primarily informed by the LF case and death data, which also shows significant spatial variability (e.g., see Redding et al. 2021) [13]. The magnitude of FOI estimates, but not their spatial distribution, varied with the seroreversion rate.

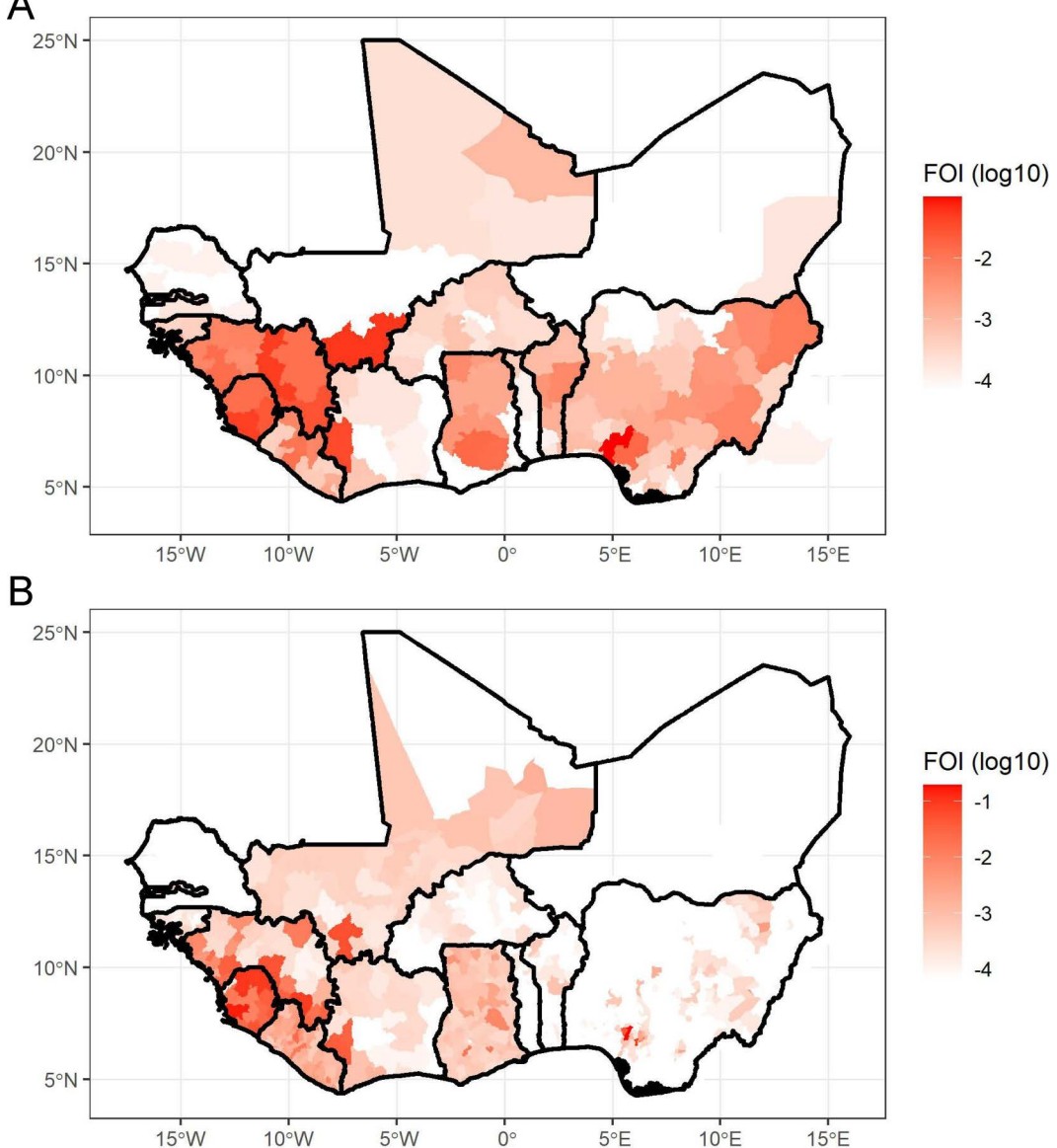

**Fig 4. Maps of FOI projections from LF case/death data and reporting probabilities at the (A) 1st and (B) 2nd administrative levels with seroreversion = 6%.** The base map layer was generated using GADM 3.6 data files which can be accessed from https://gadm.org/download_world36.html.

## Estimates of FOI from individual statistical and ensemble models

Among the statistical and machine learning models we explored to characterize the explanatory value of the covariates in S2 Table, the random forest model provided the best fit to the projected FOI estimates at the admin1 level ($r^2 = 0.95$), followed by the boosted regression model ($r^2 = 0.76$; Figs 5 and S3). At the admin2 level, the random forest model again provided the best fit ($r^2 = 0.96$), followed by the Gaussian Markov random field (GMRF) model with covariates ($r^2 = 0.80$; S4 Fig). The GMRF models with covariates fit the projected FOI estimates better than GMRF models without covariates (S3 and S4 Figs) suggesting that the covariates provide some useful information. At the admin1 level, the most important covariates in the random forest model were longitude, travel time to the nearest urban center, and the Healthcare access and quality index (HAQ) (see S3 Table for full list of covariate importance). The most influential covariates in the boosted regression model were longitude, travel time to the nearest urban center, and the occurrence of *Mastomys natalensis* (S4 Table). At the admin2 level, the most important covariates in the random forest model were longitude, HAQ, and the percentage of forest cover lost since 2000 (S5 Table). The performance of the model predictions on data held out of the regression for model validation was much lower for the 1st administrative level, with random forest providing the best fit to the testing data ($r^2 = 0.11$), suggesting that the models are overfitting to the training dataset (S5 Fig). The cross-validation performance of all of the statistical models at the 2nd administrative unit was also poor, with an $r^2 < 0.10$ for every model (S6 Fig).

Our ensemble model consisted of a weighted combination of the FOI predictions of each individual statistical regression model, along with a noise term. Due to the poor cross-validation performance of all of the statistical models, the null model received the largest weighting within the ensemble models at both the 1st and 2nd administrative level. As a result, there was limited spatial heterogeneity in the FOI predictions from the ensemble model (S7 Fig). Further details on the results of the statistical and ensemble modeling are presented in the supplement (S1 Text).

## LASV infection attack rates and LF incidence rates

Due to the positive association between the assumed seroreversion rate and FOI, the highest LASV infection attack rates and LF case incidence rates occurred in scenarios with a seroreversion rate of 6%. For a given seroreversion rate, LF case incidence rates were lowest when seropositive individuals were assumed to be protected from infection, intermediate

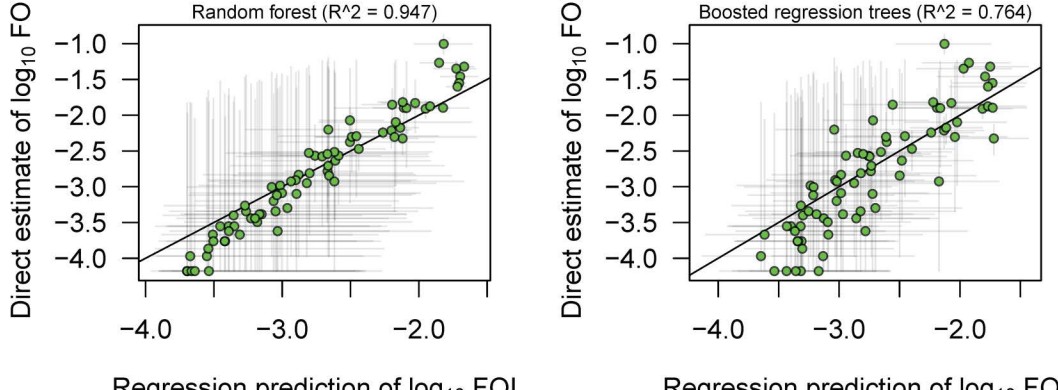

**Fig 5. Scatterplots showing the relationship between the statistical regression predictions of FOI on the x-axis versus the FOI estimates projected from LF case data and reporting probabilities for the best performing statistical models included in our analysis.** (A) Results of random forest model, and (B) results of boosted regression tree model. Plots are restricted to FOI estimates that were used in model fitting and do not include data held out for model validation. Results are for the 1st administrative level and a 6% seroreversion rate. Grey lines around points represent error bars for both direct and regression estimates of FOI values.

when they had a relative risk of 0.36 for developing LF compared to seronegative individuals, and highest when they had a relative risk of 0.53 for reinfection and disease. When seroreverted individuals were assumed to have partial protection against LF, incidence rates were lower than when seroreverted individuals were assumed to be fully susceptible. Therefore, estimated LF incidence rates were highest when we assumed that seroreversion was frequent (6%), and that both seroreverted and seropositive individuals remained susceptible to infection and disease.

Due to the spatial heterogeneity in our FOI estimates, the highest LF incidence rates were found at the admin2 as opposed to the admin1 level. No admin1 units had an LF incidence rate of greater than 10 per 1,000 (1%) with a 0% or 3% seroreversion rate, and only Ondo State in Nigeria had an LG incidence rate >10/1,000 at a 6% seroreversion rate based on our projected FOI estimates (Tables 2, S6 and S8). In general, there was a wider range of annual LF incidence rates with the projected FOI estimates than the ensemble model FOI estimates, due to the smoothing effects of the ensemble model. For example, no Nigerian states were in the top-20 using the ensemble model estimates despite Ondo, Ebonyi, and Edo States all being in the top-20 based on the projected FOI estimates. Due to the over-smoothing effect observed in the ensemble model, we focus on LF incidence rates derived from the projected FOI estimates in the rest of our results.

At the 2nd administrative level, LF incidence rates calculated from projected FOI estimates were >10 per 1,000 for several administrative units with seroreversion rates of 3% or 6%, but not 0% (Tables 3, S7 and S9). While the influence of different assumptions regarding the susceptibility of seropositive and seroreverted individuals to LF was minor at lower incidence rates, their impact is more apparent for the admin2 units with the highest incidence rates (Table 3). For example, for Ose LGA, Nigeria, the admin2 with the highest FOI, the median annual LF incidence rate increased from 9.7 per 1,000 when seropositive

**Table 2. The top 20 highest annual Lassa Fever (LF) incidence rates (per 1,000) at the 1st administrative level when the seroreversion rate is 6%. LF rates are calculated using the projected force of infection (FOI) estimates under different assumptions regarding the level of immunity in seropositive and seroreverted individuals. Values in parentheses represent 95% prediction intervals.**

| Country | Admin1 | Annual Lassa Fever incidence rate (per 1,000) | | | | | |
|---|---|---|---|---|---|---|---|
| | | Seroreverted – No Immunity | | | Seroreverted – Partial Immunity | | |
| | | Seropos. – Full Immunity | Seropos. – Part. Immunity (Hi) | Seropos. – Part. Immunity (Lo) | Seropos. – Full Immunity | Seropos. – Part. Immunity (Hi) | Seropos. – Part. Immunity (Lo) |
| Nigeria | Ondo | 9.6 (8.2 – 11.0) | 13.0 (10.3 – 16.1) | 14.6 (11.3 – 18.6) | 7.8 (6.8 – 8.7) | 11.2 (8.9 – 13.9) | 12.8 (9.9 – 16.4) |
| Mali | Sikasso | 7.0 (6.2 – 7.8) | 8.3 (7.1 – 9.5) | 8.9 (7.6 – 10.3) | 6.1 (5.4 – 6.6) | 7.3 (6.4 – 8.3) | 7.9 (6.8 – 9.1) |
| Guinea | Faranah | 6.3 (5.9 – 6.8) | 7.4 (6.8 – 8.1) | 7.9 (7.2 – 8.7) | 5.4 (5.0 – 5.7) | 6.5 (6.0 – 7.0) | 7.0 (6.4 – 7.7) |
| Sierra Leone | Southern | 6.0 (5.0– 7.1) | 7.0 (5.6 – 8.5) | 7.5 (5.9 – 9.3) | 5.2 (4.4 – 5.9) | 6.1 (5.0 – 7.4) | 6.6 (5.3 – 8.2) |
| Cote d'Ivoire | Lacs | 5.1 (3.8 – 6.4) | 5.7 (4.1 – 7.4) | 6.0 (4.3 – 8.0) | 4.4 (3.4 – 5.4) | 5.1 (3.8 – 6.5) | 5.4 (3.9 – 7.1) |
| Guinea | Nzerekore | 4.3 (4.1 – 4.5) | 4.8 (4.6 – 5.0) | 5.0 (4.8 – 5.3) | 3.9 (3.7 – 4.0) | 4.3 (4.1 – 4.5) | 4.5 (4.3 – 4.8) |
| Sierra Leone | Eastern | 4.0 (3.8 – 4.2) | 4.3 (4.1 – 4.6) | 4.5 (4.2 – 4.8) | 3.6 (3.4 – 3.7) | 3.9 (3.7 – 4.1) | 4.1 (3.9 – 4.3) |
| Nigeria | Edo | 2.6 (2.2 – 3.0) | 2.8 (2.3 – 3.2) | 2.8 (2.4 – 3.3) | 2.5 (2.1 – 2.8) | 2.6 (2.2 – 3.0) | 2.7 (2.3 – 3.1) |
| Ghana | Ashanti | 2.6 (1.1 – 4.6) | 2.7 (1.1 – 5.2) | 2.8 (1.1 – 5.4) | 2.34(1.0 – 4.0) | 2.5 (1.1 – 4.6) | 2.6 (1.1 – 4.8) |
| Sierra Leone | Western | 2.4 (1.4 – 3.7) | 2.6 (1.4 – 4.0) | 2.6 (1.4 – 4.2) | 2.3 (1.3 – 3.3) | 2.4 (1.4 – 3.7) | 2.5 (1.4 – 3.8) |
| Guinea | Kindia | 2.3 (2.1 – 2.7) | 2.4 (2.1 – 2.8) | 2.5 (2.2 – 2.9) | 2.2 (1.9 – 2.5) | 2.3 (2.0 – 2.6) | 2.4 (2.1 – 2.7) |
| Sierra Leone | Northern | 2.2 (2.1 – 2.4) | 2.3 (2.2 – 2.5) | 2.4 (2.2 – 2.6) | 2.1 (2.0 – 2.3) | 2.2 (2.1 – 2.4) | 2.3 (2.1 – 2.4) |
| Ghana | Eastern | 2.2 (1.2 – 3.7) | 2.3 (1.2 – 4.0) | 2.4 (1.2 – 4.2) | 2.1 (1.1 – 3.3) | 2.2 (1.2 – 3.6) | 2.2 (1.2 – 3.8) |
| Guinea | Kankan | 2.2 (1.7 – 2.7) | 2.3 (1.7 – 2.9) | 2.3 (1.8 – 3.0) | 2.0 (1.6 – 2.5) | 2.1 (1.6 – 2.7) | 2.2 (1.7 – 2.8) |
| Liberia | Grand Bassa | 2.2 (0.4 – 6.9) | 2.3 (0.4 – 8.4) | 2.4 (0.4 – 9.1) | 2.1 (0.4 – 5.8) | 2.2 (0.4 – 7.2) | 2.2 (0.4 – 7.9) |
| Nigeria | Borno | 1.6 (1.0 – 2.2) | 1.6 (1.0 – 2.3) | 1.6 (1.0 – 2.4) | 1.5 (1.0 – 2.1) | 1.5 (1.0 – 2.2) | 1.6 (1.0 – 2.3) |
| Liberia | Bong | 1.5 (0.2 – 7.1) | 1.5 (0.2 – 8.7) | 1.5 (0.2 – 9.4) | 1.4 (0.2 – 5.9) | 1.5 (0.2 – 7.4) | 1.5 (0.2 – 8.2) |
| Guinea | Labe | 1.2 (0.8 – 1.9) | 1.3 (0.8 – 1.9) | 1.3 (0.8 – 2.0) | 1.2 (0.7 – 1.8) | 1.2 (0.8 – 1.8) | 1.3 (0.8 – 1.9) |
| Nigeria | Ebonyi | 1.3 (0.1 – 7.6) | 1.2 (0.1 – 9.3) | 1.2 (0.1 – 10.0) | 1.1 (0.1 – 6.3) | 1.2 (0.1 – 8.0) | 1.2 (0.1 – 8.8) |
| Guinea | Boke | 1.1 (0.7 – 1.7) | 1.2 (0.7 – 1.8) | 1.2 (0.7 – 1.8) | 1.1 (0.7 – 1.6) | 1.1 (0.7 – 1.7) | 1.1 (0.7 – 1.7) |

**Table 3. The top 20 highest annual Lassa Fever (LF) incidence rates (per 1,000) at the 2ⁿᵈ administrative level when the seroreversion rate is 6%.** LF rates are calculated using the projected force of infection (FOI) estimates under different assumptions regarding the level of immunity in seropositive and seroreverted individuals. Values in parentheses represent 95% prediction intervals.

| Country | Admin1 | Admin2 | Annual Lassa Fever incidence rate (per 1,000) | | | | | |
|---|---|---|---|---|---|---|---|---|
| | | | Seroreverted – No Immunity | | | Seroreverted – Partial Immunity | | |
| | | | Seropos. – Full Immunity | Seropos. – Part. Immunity (Hi) | Seropos. – Part. Immunity (Lo) | Seropos. – Full Immunity | Seropos. – Part. Immunity (Hi) | Seropos. – Part. Immunity (Lo) |
| Nigeria | Ondo | Ose | 12.4 (10.1 – 13.8) | 20.7 (14.1 –28.2) | 24.7 (15.9 – 35.2) | 9.7 (8.1 – 10.6) | 18.0 (12.1 – 25.1) | 22.0 (14.1 – 32.0) |
| Sierra Leone | Southern | Moyamba | 11.2 (9.5 – 12.5) | 17.6 (13.0 – 22.7) | 20.7 (14.7 – 27.6) | 8.7 (7.6 – 9.5) | 15.0 (11.1 – 19.8) | 18.2 (12.7 – 24.7) |
| Nigeria | Edo | Esan West | 9.7 (8.3 – 10.9) | 13.2 (10.5 – 16.0) | 14.9 (11.4 – 18.5) | 7.9 (6.8 – 8.7) | 11.3 (9.0 – 13.8) | 13.0 (10.0 – 16.2) |
| Sierra Leone | Northern | Bombali | 9.5 (5.8 – 12.2) | 13.1 (6.7 – 21.4) | 14.9 (7.2 – 25.9) | 7.6 (5.0 – 9.3) | 11.2 (5.8 – 18.6) | 12.9 (6.3 – 23.0) |
| Sierra Leone | Northern | Koinadugu | 9.1 (2.2 -13.1) | 12.2 (2.3 – 27.0) | 13.8 (2.4 – 33.6) | 7.3 (2.1 – 9.9) | 10.4 (2.1 – 23.8) | 12.0 (2.2 – 30.6) |
| Guinea | Nzerekore | Macenta | 9.0 (8.2 – 9.6) | 11.8 (10.5 – 13.3) | 13.2 (11.6 – 15.0) | 7.2 (6.7 – 7.7) | 10.1 (9.0 – 11.3) | 11.5 (10.0 – 13.0) |
| Mali | Sikasso | Bougouni | 7.0 (6.3 – 7.8) | 8.3 (7.3 –9.4) | 8.9 (7.8 – 10.2) | 6.1 (5.5 – 6.6) | 7.3 (6.5 – 8.3) | 8.0 (6.9 – 9.1) |
| Guinea | Faranah | Faranah | 6.3 (5.8 – 6.8) | 7.4 (6.8 – 8.1) | 8.0 (7.2 – 8.8) | 5.4 (5.0 – 5.7) | 6.5 (5.9 – 7.1) | 7.0 (6.4 – 7.7) |
| Sierra Leone | Southern | Pujehun | 6.3 (2.0 – 10.7) | 7.4 (2.1 – 15.9) | 7.9 (2.1 – 18.4) | 5.4 (1.9 – 8.3) | 6.5 (2.0 – 13.5) | 7.0 (2.1 – 16.1) |
| Nigeria | Ondo | Owo | 5.9 (3.6 – 8.2) | 6.7 (3.9 – 10.2) | 7.1 (4.0 – 11.2) | 5.1 (3.3 – 6.7) | 5.9 (3.5 – 8.8) | 6.4 (3.6 – 9.8) |
| Sierra Leone | Eastern | Kailahun | 5.6 (3.4 – 7.8) | 6.4 (3.7 – 9.6) | 6.8 (3.8 – 10.6) | 4.8 (3.1 – 6.4) | 5.6 (3.4 – 8.3) | 6.0 (3.5 – 9.2) |
| Cote d'Ivoire | Montagnes | Cavally | 5.3 (3.7 – 6.9) | 6.0 (4.1 – 8.3) | 6.4 (4.2 – 9.0) | 4.6 (3.4 – 5.8) | 5.3 (3.7 – 7.2) | 5.7 (3.9 – 7.9) |
| Sierra Leone | Eastern | Kono | 5.1 (0.2 – 12.1) | 5.7 (0.2 – 20.9) | 6.0 (0.2 – 25.2) | 4.5 (0.2 – 0.2) | 5.1 (0.2 – 18.0) | 5.4 (0.2 – 22.5) |
| Sierra Leone | Western | Western Rural | 5.0 (0.1 – 12.2) | 5.7 (0.1 – 21.0) | 5.9 (0.1 – 25.5) | 4.4 (0.1 – 9.3) | 5.0 (0.1 – 18.2) | 5.3 (0.1 – 22.4) |
| Guinea | Kindia | Kindia | 4.9 (4.0 – 5.9) | 5.5 (4.4 – 6.8) | 5.8 (4.5 – 7.2) | 4.3 (3.6 – 5.1) | 4.9 (3.9 – 5.9) | 5.2 (4.1 – 6.4) |
| Guinea | Nzerekore | Gueckedou | 4.6 (4.3 – 5.0) | 5.1 (4.7 – 5.6) | 5.4 (4.9 – 5.9) | 4.1 (3.8 – 4.4) | 4.6 (4.3 – 5.0) | 4.9 (4.5 – 5.3) |
| Guinea | Nzerekore | Lola | 4.4 (3.8 – 5.1) | 4.9 (4.2 – 5.7) | 5.1 (4.3 – 6.0) | 3.9 (3.5 – 4.4) | 4.4 (3.8 – 5.1) | 4.6 (3.9 – 5.4) |
| Cote d'Ivoire | Montagnes | Guemon | 4.4 (2.3 – 6.9) | 4.8 (2.4 – 8.3) | 5.1 (2.4 – 8.9) | 3.9 (2.2 – 5.8) | 4.4 (2.2 – 7.2) | 4.6 (2.3 – 7.9) |
| Guinea | Nzerekore | Yamou | 4.2 (3.7 – 4.7) | 4.6 (4.0 – 5.3) | 4.8 (4.2 – 5.5) | 3.8 (3.4 – 4.2) | 4.2 (3.7 – 4.7) | 4.4 (3.8 – 4.9) |
| Sierra Leone | Eastern | Kenema | 3.9 (3.7 – 4.1) | 4.3 (4.0 – 4.5) | 4.5 (4.2 – 4.7) | 3.5 (3.4 – 3.7) | 3.9 (3.7 – 4.1) | 4.1 (3.8 – 4.3) |

individuals had full immunity and seroreverted individuals had partial protection, to 12.4 per 1,000 when seropositive individuals were fully immune but seroreverted individuals had no protection, to 24.7 per 1,000 when seropositive individuals were only partially immune and seroreverted individuals had no protection. Four admin2 units (two in Sierra Leone, one in Nigeria, and one in Guinea) had LF incidence rates <10 per 1,000 when seropositive individuals were fully protected, but incidence rates above 10 per 1,000 when we assume seropositive individuals were susceptible to reinfection and illness.

The age-specific infection histories calculated using the projected and ensemble FOI estimates can also be used to calculate annual LF incidence rates for specific age groups. LF incidence rates decrease with age as the likelihood of a previous infection and at least partial protection from infection and disease increases (Fig 6).

## Interannual variability in incidence

The 1,000 samples from the posterior distribution of the projected FOI estimates incorporate both the uncertainty and the interannual variability in our FOI estimates and the corresponding annual LF incidence rates. Fig 7 provides an example

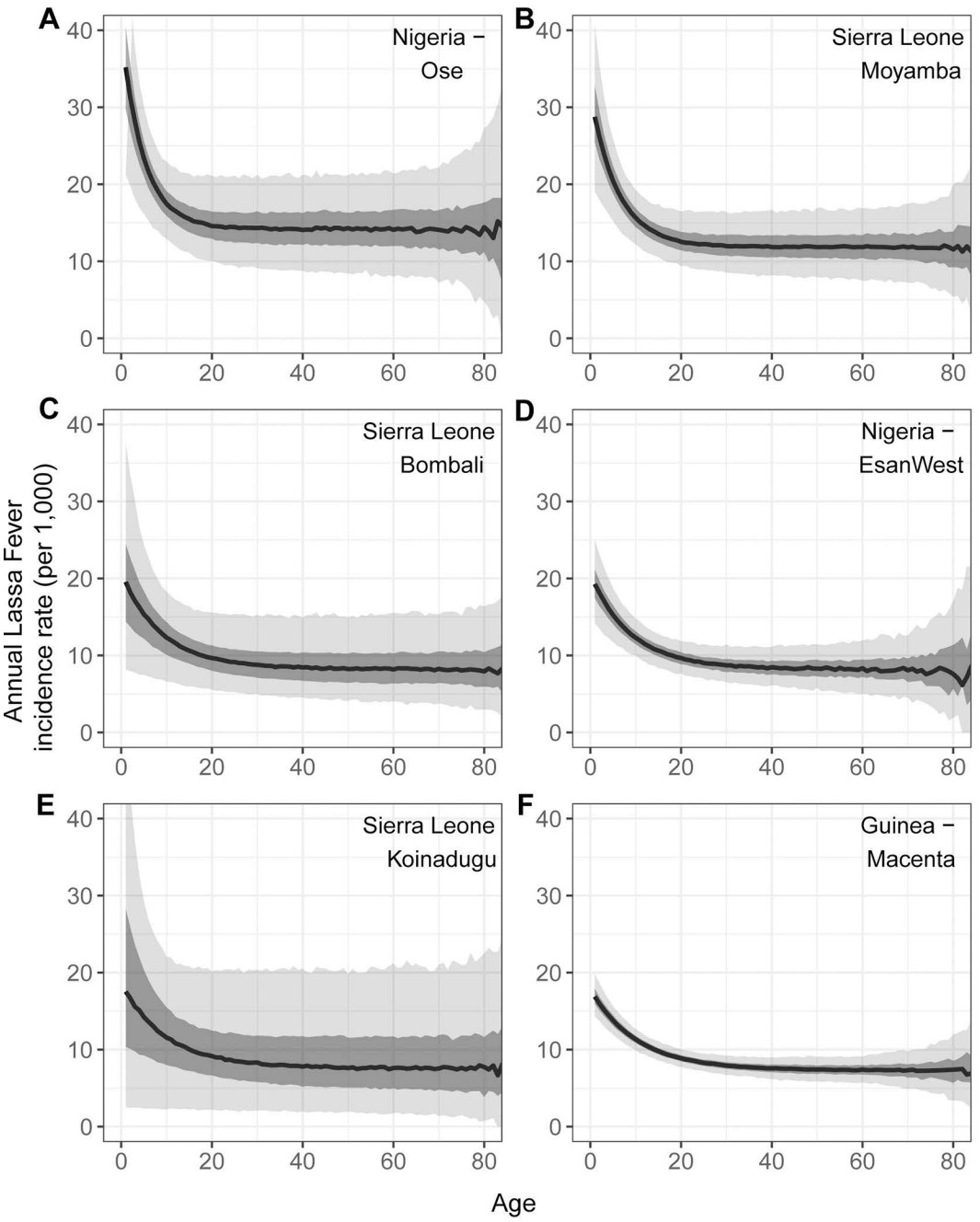

**Fig 6. Median annual Lassa Fever (LF) incidence rates per 1,000 by age in the six 2ⁿᵈ administrative units with the highest incidence.** Incidence calculated using projected FOI estimates from LF case data and estimated reporting probabilities and assuming a seroreversion rate of 6%, seropositive individuals have partial protection from reinfection and disease (RR = 0.53), and seroreverted individuals are partially protected (RR = 0.36) from reinfections and disease. Dark grey represents the interquartile range (IQR) and lighter grey the 95% prediction interval.

of the variability in LF incidence rates (per 1,000) for the nine highest incidence admin2 units under a scenario where the seroreversion rate is 6%, seroprotected individuals have a relative risk = 0.36 of developing LF if infected, and seroreverted individuals are also partially susceptible to reinfection and disease (relative risk = 0.53). The variation in LF incidence

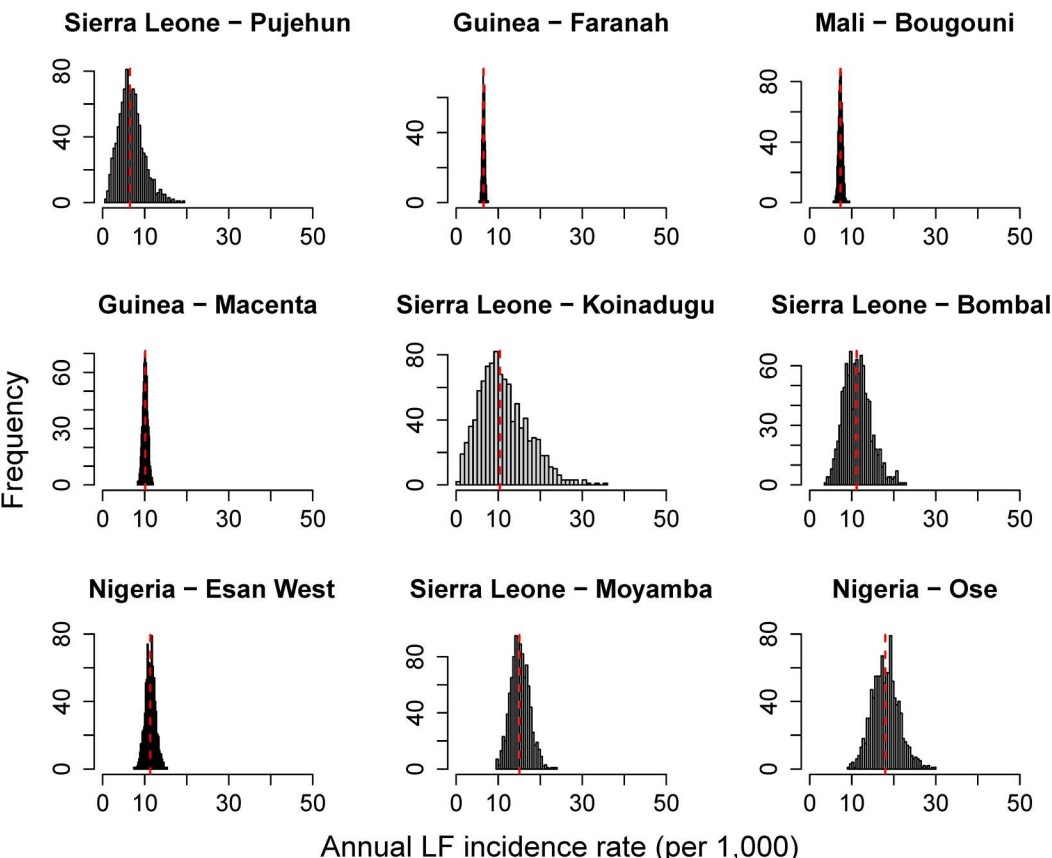

**Fig 7. Posterior distribution of annual Lassa fever (LF) case incidence rates in the nine highest admin2 units.** LF incidence estimates are based on projected FOI estimates from LF case data and reporting probabilities. Results presented are for a scenario with 6% seroreversion rate, partial protection against reinfection and disease in seropositive individuals (relative risk = 0.36), and partial protection against reinfection or disease among seroreverted individuals (RR = 0.53). The red dashed line is the median of the posterior distribution.

rates is highest for locations where FOI was estimated from LF case data and reporting probabilities only (no serology data) because these estimates incorporate uncertainty in reporting probabilities in addition to interannual variability. Locations where FOI estimates were informed by serology data, such as Macenta and Faranah Districts in Guinea and Sikasso District in Mali, have lower uncertainty. However, even in these locations, the estimated annual LF incidence rate can vary by 10–50% from year-to-year. For example, the narrowest estimated range in annual LF incidence is in Faranah District, Guinea with a median annual incidence of 7.0 per 1,000 (95% CrI: 6.4-7.7), where 95% of years would be expected to be within +/- 10% of the median value. Esan West LGA in Edo State, Nigeria has a median LF incidence rate of 13.0 per 1,000 (95% CrI: 10.0-16.2), with variability of +/- 25% from the median. An example of a location with a high uncertainty and interannual variability is Ose LGA in Ondo State, Nigeria which has a median annual LF incidence rate of 22.0 per 1,000 (95% CrI: 14.1-32.0) with variability of approximately +/- 50%.

## Seasonality

Reported LF cases in Nigeria and Liberia, and to a lesser extent in Sierra Leone, show a clear seasonal pattern with a peak in cases in January to March (S8 Fig). Liberia also shows a secondary peak later in the year, although this may be part of the January peak in cases. In Sierra Leone, LF cases peak in March, but there appears to be considerable transmission throughout the year.

## Discussion

Using a modeling framework that incorporated LF serology, case, and death data, we found considerable spatial variation in LASV spillover and LF incidence across West Africa, with the highest incidence rates in areas within Nigeria, Sierra Leone, and Guinea. We also estimate that as few as 0.2% of LASV infections are captured by current surveillance systems. Our LF incidence estimates were sensitive to assumptions about the duration and strength of infection-induced immunity. LF incidence rates were particularly sensitive to the rate of seroreversion among previously infected individuals because this value affects both susceptibility to reinfection and the interpretation of serology data. Our spatial LF incidence rate estimates, along with the interannual and seasonal variability in these estimates, could be used to target high incidence areas suitable for inclusion in a vaccine trial and estimate expected trial event rates. However, the uncertainties in our LF incidence estimates highlight critical knowledge gaps regarding the number of asymptomatic and mild LASV infections that go undetected and the extent to which these infections provide long-lasting immunity.

Our estimates of LF incidence rates indicate that there are few 1st or 2nd level administrative districts where the predicted attack rate would be at least 1% as is desired for vaccine field trials. Our estimates assumed that 20% of LASV infections are symptomatic, as frequently reported. However, if substantially fewer than 20% of infections are captured by active syndromic surveillance, as has initially been reported for the Enable study, then none of these districts would be likely to reach an LF attack rate of 1% [23]. Therefore, ensuring sufficient statistical power may require a large increase in the number of individuals enrolled in a field trial or a longer trial period spanning multiple transmission seasons. One alternative to increasing the size of the study population would be to use protection against infection as a primary endpoint instead of protection against symptomatic disease. Due to the high number of asymptomatic infections, active monitoring for seroconversion—while more difficult and costly than symptom-based surveillance methods—would increase the number of expected endpoints without increasing the size of the study population. Another option would be to adopt a responsive trial design that employed ring vaccination or a similar strategy to focus study efforts on locations with active transmission. Given the focal nature of LF spillover to humans, and the substantial interannual and seasonal variation in incidence, such a strategy would ensure that areas of active transmission are targeted [4,47,58].

Trial site selection also needs to account for the baseline seroprevalence in a target population, as that will influence the fraction of the population that is susceptible to infection. Locations with high baseline seroprevalence may experience few LF cases even if LASV is actively circulating in the rodent population. However, there is considerable uncertainty about the duration of immunity to LASV, and several studies suggest that seroreversion is relatively common [4,14,59]. Assumptions about the seroreversion rate had the largest impact on estimated LF incidence among the different immunological scenarios included in our analysis. Without seroreversion, our FOI estimates were too low for the resulting annual LF incidence rates to exceed 10 per 1,000 (1%) anywhere within the study region. Further results from the longitudinal serology samples from the Enable study should help refine our understanding of seroreversion rates and whether they vary by location or age.

The level of protection against reinfection and disease among both seropositive and seroreverted individuals also influenced expected LF incidence rates in our model, with higher levels of protection against disease resulting in lower expected incidence rates. The different scenarios explored in our model could be leveraged to select the most plausible scenario for estimating event rates in a particular site and target population. This model can also be used to explore how LF incidence rates vary by both age and serostatus under different assumptions regarding how serostatus affects susceptibility to (re-)infection and disease, which can help to inform selection of a target population and trial size calculations. Results from the Enable study will also help refine future model scenarios regarding the role of immunity and serostatus, as the study will report relative LF incidence rates among individuals who were seropositive or seronegative at baseline (or in the previous sampling period) and may also be able to capture the reinfection and LF incidence rates among individuals who serorevert over the course of the study.

Our model projections represent the most extensive and geographically detailed estimates of LF surveillance and incidence across the entire endemic range in West Africa to date. Previous modeling studies have generated fine-scale

maps of the likely distribution of LASV, but have not estimated LF incidence rates or seroprevalence in the human population [60,61,62]. Basinski et al. (2021) modeled LASV risk in rodents and then fit a regression model of this risk measure against historical seroprevalence data to generate fine-scale estimates of LASV seroprevalence in the human population [5]. However, their study did not incorporate LF incidence data or account for certain epidemiological features of LF in generating these estimates. Our projected FOI estimates and modeled annual LF incidence rates indicate that Sierra Leone, southern Guinea near the border with Sierra Leone and Liberia, and a few high incidence regions within Nigeria would likely yield the highest LF case incidence rates during a vaccine trial. Comparisons of our estimates at the 1st and 2nd administrative levels show that there is considerable spatial heterogeneity among different admin2s within the same 1st administrative unit (particularly in Nigeria, Guinea, and Mali), and therefore predictions from the 2nd administrative level are likely to be more useful for site selection.

The 2nd administrative level FOI estimates that were projected from LF case data and reporting probabilities appear to be more accurate than the FOI estimates from our ensemble model that used spatial covariates to improve model predictions in areas lacking data, particularly in areas that are projected to have the highest incidence. None of the individual statistical models within the ensemble adequately predicted the test data held out from the initial model fitting process and the ensemble model smoothed over the spatial heterogeneity in FOI to an extent that lowered the incidence rate in some regions with a high number of reported cases and deaths, particularly in Edo and Ondo states in Nigeria. Therefore, at present, the projected FOI estimates provide more reliable estimates for field trial site selection than the estimates from the ensemble model. Further model refinements, including model selection techniques to determine the most influential spatial covariates, and further model validation using serology data from the literature and forthcoming Enable results could improve the ensemble model predictions. However, regression analyses and ensemble modeling methods are most useful for predicting incidence where data is sparse and the relationship between the response variable (incidence) and the explanatory variables is strong. There is still a lot of uncertainty about what conditions distinguish areas with high LASV spillover rates from areas with similar environmental conditions and where *Mastomys spp.* are present, but spillover is rare or nonexistent. Hopefully additional field studies and serological surveys will help explain these discrepancies, but at present we lack the ability to predict the occurrence of LF at a fine spatial scale outside of the well-documented hotspots of transmission. In the absence of this ability, our admin2 incidence estimates could help identify broader regions to target for vaccine trials, and baseline serology surveys can be conducted at the local level to confirm LASV spillover in the area. In particular, evidence of past infection in younger children would indicate recent transmission.

Our modeling framework did incorporate estimation of country specific LF case and death reporting probabilities, but the extent to which LF hotspots might exist in areas that haven't reported any LF cases or deaths is unknown. Current hotspots are associated with the presence of LF surveillance systems and the locations of specialty diagnostic and treatment centers, suggesting that transmission may be occurring undetected where these systems are lacking. For example, high seroprevalence was recently observed in southern Mali, suggesting that there may be undocumented areas of LF incidence outside of the historical hotspots [63,64]. Although the burden of LF outside of the known hotspots is an important outstanding question, there is some evidence that observed spatial patterns of reported LF cases and deaths reflect at least some important differences in the spatial distribution of the disease. Although mild and moderate LF cases are difficult to distinguish from other febrile illnesses such as malaria, severe LF cases requiring hospitalization have been associated with nosocomial outbreaks in the known hotspots in Nigeria, Sierra Leone, and Liberia, but not in other areas of West Africa [65–67]. A seroprevalence study in multiple locations within Ghana, and preliminary seroprevalence results from the Enable site in Benin also suggest that transmission is lower in these countries than it is in the known hotspots for transmission [68].

## Study limitations

Despite recent efforts to prioritize the study of LF, there are still many unknowns, which limit the predictive power of our model. LF incidence rates vary significantly based on many confluent factors, and the limited number of longitudinal and

broad scale studies makes it difficult to draw significant conclusions about the risk of LF in a particular time and place. The epidemiology of LF may have also changed over the course of the period covered in our datasets, which might bias our results if spillover rates have significantly increased or decreased in some regions. We assumed a constant rodent-to-human FOI because we did not have sufficient data to detect temporal shifts in FOI. The case and death data contained limited phylogenetic information, such as the LASV lineage, that might affect disease severity, and therefore the probability of a case being detected. There is also variability in the sensitivity and specificity of the different serological tests used in the studies included in our analysis, as well as variability in the LF case definitions used in different countries, which could lead to temporal or spatial biases in case surveillance. Our model does not account for non-epidemiological considerations that may influence site selection, such as the strength of the existing local or national health infrastructure system, political stability, or cultural barriers to trial implementation. However, the model results can be used to rank potential trial sites by expected LF incidence rates and seroprevalence (e.g., expected serostatus by age group), and then other factors can be used to select appropriate trial sites from locations with suitable characteristics.

The current model also cannot estimate the geographical variation in expected LF incidence rates within a given 2nd administrative area. Past serology studies in Guinea and Sierra Leone, and preliminary results from the Enable study, show that seroprevalence rates can vary significantly from village to village within the same state or district [4,47,58]. However, the identification of risk factors associated with small-scale variations in seroprevalence, infection attack rates, or LF incidence have been inconsistent, limiting our ability to predict high incidence areas within a given administrative region. The Enable study will provide some additional context on this finer-scale heterogeneity in attack rates and incidence due to the large sample size and the relatively high number of villages sampled. However, the targeted criteria used for site selection may limit our ability to extrapolate the study results beyond these study sites. Variation in our estimated LF incidence rates for a given location results from a combination of parameter uncertainty and interannual variability in reported LF cases and deaths. Longitudinal serological or incidence data was insufficient to explicitly estimate interannual variability across the study area.

## Conclusions

Our modeling framework enabled us to leverage multiple data sources to estimate LF incidence at the 2nd administrative level across West Africa. Expected incidence varied considerably and showed marked geographic variation across spatial scales. Although an ensemble of regression models showed moderate success at predicted incidence based on environmental and socioeconomic data, it tended to underestimate incidence in high-risk regions, which are the most relevant for disease control and vaccine trial planning. Our work highlights the importance of ecological and immunological factors and underscores large uncertainties in our understanding of LF epidemiology. Our findings emphasize the need for more prospective data (e.g., the Enable study), particularly regarding the fraction of infections that are detectable by syndromic surveillance and the duration of infection-induced immunity. At the scales considered, very few locations in West Africa are predicted to experience LF incidence at the levels needed to conduct a vaccine efficacy trial (annual incidence of at least 10 per 1000). Designs would need to accommodate low disease incidence (e.g., preparatory observational studies and active monitoring for asymptomatic infections) or look to prospectively enrich the trial population with at risk individuals (e.g., ring vaccination). Our modeling framework is designed to be updated iteratively with future serological survey data and LF surveillance data. Our estimated incidence rates are intended to assist with trial site selection, sample size calculations, and the decisions regarding the appropriate target population and primary endpoint.

## Supporting information

**S1 Text. Model details and supplemental results.**
(DOCX)

**S1 Table. Papers selected for in-depth literature review.**
(XLSX)

**S2 Table. Population-level covariates with a potential association with LF occurrence.**
(XLSX)

**S3 Table. Variable importance in the Random Forest model at the 1st administrative level.** Variable importance calculated with median projected FOI as the response variable.
(XLSX)

**S4 Table. Variable importance in the Boosted regression tree model at the 1st administrative level.** Variable importance calculated with median projected FOI as the response variable.
(XLSX)

**S5 Table. Variable importance in the Random Forest model at the 2nd administrative level.** Variable importance calculated with median projected FOI as the response variable.
(XLSX)

**S6 Table. The top 20 highest annual Lassa Fever (LF) incidence rates (per 1,000) at the 1st administrative level when the seroreversion rate is 0%.**
(XLSX)

**S7 Table. The top 20 highest annual Lassa Fever (LF) incidence rates (per 1,000) at the 2nd administrative level when the seroreversion rate is 0%.**
(XLSX)

**S8 Table. The top 20 highest annual Lassa Fever (LF) incidence rates (per 1,000) at the 1st administrative level when the seroreversion rate is 3%.**
(XLSX)

**S9 Table. The top 20 highest annual Lassa Fever (LF) incidence rates (per 1,000) at the 2nd administrative level when the seroreversion rate is 3%.**
(XLSX)

**S1 Fig.** Maps of FOI projections from LF case/death data and reporting probabilities at the (A) 1st and (B) 2nd administrative levels with seroreversion = 0%. The base map layer was generated using GADM 3.6 data files which can be accessed from https://gadm.org/download_world36.html
(TIF)

**S2 Fig.** Maps of FOI projections from LF case/death data and reporting probabilities at the (A) 1st and (B) 2nd administrative levels with seroreversion = 3%. The base map layer was generated using GADM 3.6 data files which can be accessed from https://gadm.org/download_world36.html
(TIF)

**S3 Fig. Scatterplots showing the relationship between the statistical regression predictions of FOI on the x-axis vs. the FOI estimates projected from LF case data and reporting probabilities for each of the eight statistical models (plus a null intercept-only model) included in our analysis.**
(TIFF)

**S4 Fig. Scatterplots showing the relationship between the statistical regression predictions of FOI on the x-axis vs. the FOI estimates projected from LF case data and reporting probabilities for each of the eight statistical models (plus a null intercept-only model) included in our analysis.**
(TIFF)

**S5 Fig. Scatterplots showing the cross-validation performance of each statistical regression model.**
(TIFF)

**S6 Fig. Scatterplots showing the cross-validation performance of each statistical regression model.**
(TIFF)

**S7 Fig. Map of ensemble model-based FOI estimates at the (A) 1st and (B) 2nd administrative levels with seroreversion = 6%.** The base map layer was generated using GADM 3.6 data files which can be accessed from https://gadm.org/download_world36.html
(TIF)

**S8 Fig. Seasonality of reported LF cases in (A) Nigeria, (B) Liberia, (C) Sierra Leone, and (D) averaged across these three countries.**
(TIF)

## Acknowledgments

The authors thank CEPI for insights into current planning considerations for Lassa fever vaccine trials and an overview of the preliminary results from the ENABLE study. We thank Melissa Wynn, Carrie Mills, Kevin Sprurgers, and Lovelyn Charles at Emergent BioSolutions for help with coordinating research efforts.

## Author contributions

**Conceptualization:** Sean M. Moore, Natalie E Dean, Steven T. Stoddard.

**Data curation:** Sean M. Moore, Erica Rapheal, Sandra Mendoza Guerrero.

**Formal analysis:** Sean M. Moore.

**Funding acquisition:** Steven T. Stoddard.

**Investigation:** Sean M. Moore.

**Methodology:** Sean M. Moore, Natalie E Dean, Steven T. Stoddard.

**Project administration:** Sandra Mendoza Guerrero, Steven T. Stoddard.

**Resources:** Sean M. Moore.

**Software:** Sean M. Moore.

**Supervision:** Steven T. Stoddard.

**Validation:** Sean M. Moore.

**Visualization:** Sean M. Moore, Erica Rapheal.

**Writing – original draft:** Sean M. Moore, Erica Rapheal.

**Writing – review & editing:** Sean M. Moore, Erica Rapheal, Sandra Mendoza Guerrero, Natalie E Dean, Steven T. Stoddard.

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
