## [Decision Letter · Decision Letter 0]

Response to Reviewers
Revised Manuscript with Track Changes
Manuscript

Ran Wang, M.D.

Academic Editor

Shaden Kamhawi

co-Editor-in-Chief

Paul Brindley

co-Editor-in-Chief

**Journal Requirements:**

At this stage, the following Authors/Authors require contributions: Sean M. Moore, Erica Rapheal, Sandra Mendoza Guerrero, Natalie E Dean, and Steven T. Stoddard. Please ensure that the full contributions of each author are acknowledged in the "Add/Edit/Remove Authors" section of our submission form.

2) We noticed that you used the phrase 'unpublished data' in the manuscript. We do not allow these references, as the PLOS data access policy requires that all data be either published with the manuscript or made available in a publicly accessible database. Please amend the supplementary material to include the referenced data or remove the references.

Potential Copyright Issues:

- Figures 1, 3, 4, S1, S2, S7, S8, and S9; Please provide a direct link to the base layer of the map (i.e., the country or region border shape) and ensure this is also included in the figure legend; and provide a link to the terms of use / license information for the base layer image or shapefile. We cannot publish proprietary or copyrighted maps (e.g. Google Maps, Mapquest) and the terms of use for your map base layer must be compatible with our CC BY 4.0 license.

**Reviewers' comments:**

**Key Review Criteria Required for Acceptance?**

**Methods**

-Are the objectives of the study clearly articulated with a clear testable hypothesis stated?

-Is the study design appropriate to address the stated objectives?

-Is the population clearly described and appropriate for the hypothesis being tested?

-Is the sample size sufficient to ensure adequate power to address the hypothesis being tested?

-Were correct statistical analysis used to support conclusions?

-Are there concerns about ethical or regulatory requirements being met?

Reviewer #1: This is a very important and relevant Manuscript especially with the prioritization of Lassa as a disease needing Countermeasures. The objectives are clear, smart and addresses a critical issue in Lassa research. The number of countries and sub nationals included for the modelling are quite appropriate. The Statistical analysis used to support the conclusions are also appropriate

Reviewer #2: The objectives of the study were clearly articulated and the study design was appropriate. To the best of my knowledge, the correct statistical analyses were used.

Reviewer #3: The manuscript frequently highlights substantial differences in observed seroprevalence at the village level, even within the same administrative level 2 region. This suggests that aggregating FOI at the chosen level of analysis may overlook critical heterogeneities. While the framing of the study as identifying potential vaccine trial sites at administrative levels 1 or 2 partly justifies this choice, the rationale for selecting this level of aggregation could be made more explicit. Specifically, the manuscript would benefit from a discussion of why a pixel-based approach was not used. Such an approach might better capture intra-regional variability, reduce the impact of administrative unit size on results, and more effectively represent heterogeneity in the included covariates. I am not asking for the analysis to be refactored at this level but I do believe a discussion of consideration of this would be suitable.

See summary and general comments for minor revisions suggested for the methods.

**Results**

-Does the analysis presented match the analysis plan?

-Are the results clearly and completely presented?

-Are the figures (Tables, Images) of sufficient quality for clarity?

Reviewer #1: The Analysis presented are satisfactory and match the plan. The results are clearly and completely presented. I find it important that the Authors understand that seroreversion does not necessarily equate to loss of protective immunity.

Reviewer #2: The results were clearly presented and consistent with the stated analytical plan. Figures and tables were excellent.

Reviewer #3: See summary and general comments for minor revisions suggested for the results.

**Conclusions**

-Are the conclusions supported by the data presented?

-Are the limitations of analysis clearly described?

-Do the authors discuss how these data can be helpful to advance our understanding of the topic under study?

-Is public health relevance addressed?

Reviewer #1: The conclusions are well made and supported by the data provided. The Authors recognize and have clearly stated the study limitation. These are due to many uncertainties about Lassa virus epidemiology

Reviewer #2: The conclusions were appropriate. Limitations could be enhanced (see below).

Reviewer #3: See summary and general comments for minor revisions suggested for the discussion/conclusion.

**Editorial and Data Presentation Modifications?**

Reviewer #1: ACCEPT

Reviewer #2: See comments below.

Reviewer #3: See summary and general comments.

**Summary and General Comments**

Reviewer #1: This is a very strong study that will not only contribute to knowledge but will likely have immediate applicability in Clinical trial design for Lassa vaccine and Lassa therapeutic trials.

Reviewer #2: In their manuscript entitled, “Estimation of Lassa Fever Incidence Rates in West Africa: Development of a Modeling Framework to Inform Vaccine Trial Design”, Moore and colleagues determined the estimated force of infection of Lassa fever in administrative units in West Africa using previously published seroprevalence data as well as LF case and death reports. 31 published manuscripts were selected for review. Based on available serologic data, they were able to calculate the FOI for 24 1st level administrative units (Provinces/States) and 53 2nd administrative units (chiefdoms/municipalities). Assuming no seroreversion, Ondo State in Nigeria had the highest FOI at the Admin1 level and Moyamba District, Sierra Leone had the greatest FOI at the admin2 level. When seroreversion was accounted for, Ondo State still had the highest Admin1 FOI while OseLGA, Nigeria had the highest Admin2 rate. The random forest model proved to have the best fit among the statistical and machine learning models tested. Here, at the admin1 level, the most important covariates were longitude, travel time to nearest urban center and the HAQ. At the admin2 level, the most important covariates were longitude, HAQ and precipitation.

Lassa fever remains a significant health burden in West Africa, especially in the countries of Guinea, Sierra Leone, Liberia and Nigeria. There is increasing concern about cases appearing in other countries such as Benin and Togo. No vaccine is currently available, but several are in phase 1 trials and phase 2 trials are scheduled to begin soon. As the authors note, a greater understanding of the epidemiology of Lassa fever is needed to identify areas of West Africa that are best suited for vaccine trials. The design and conduct of these trials is complicated by the fact that an estimated 20% of infections result in disease severe enough to be recognized. Therefore, it has been estimated that a minimum annual incidence rate of 1% is needed to adequately power a vaccine efficacy trial. Unfortunately, there is a lack of prospective epidemiologic studies. In addition, there are several other factors which make determining the FOI accurately. First, and most importantly, this study is based on previously published seroprevalence studies that used very different diagnostic testing to determine sero-positivity. These differences make it very challenging to compare studies. Second, sero-reversion has been noted in many incidence studies. The authors accounted for these issues and others to develop 18 different models. Although I am not a statistician and their methods are beyond my area of expertise, the approach seems very thorough.

Overall, this is a well written manuscript that will increase our understanding of Lassa virus epidemiology. In addition, the results challenge commonly held notions about LF incidence. For example, in Sierra Leone, the districts of Kenema and Kailahun have traditionally been considered LF hyperendemic areas. The results described in this manuscript challenge these beliefs and may result in improved design of vaccine trials and the identification of areas most at risk and, consequently, the populations that would most benefit from a vaccine.

Specific Comments:

1. Line 396: This line has an incomplete sentence that must be revised.

2. Discussion: The section on limitations neglects several key factors. First, it is certainly possible that LF epidemiology has changed over the past 20 years. Including all studies equally may bias the results. Second, virtually all studies used a different platform for determining seropositivity. They may not be comparable. Third, Lassa virus is now considered to be made up of as many as 7 different lineages. It is conceivable that not all lineages are equally likely to cause infection.

Reviewer #3: Review of - Estimation of Lassa fever incidence rates in West Africa: development of a modeling framework to inform vaccine trial design

Thank you for the opportunity to review this manuscript which investigates the force of infection (FOI) of Lassa virus (LASV) in endemic regions of West Africa. The authors aim to model the spatially explicit FOI across West Africa using serological survey data and published case data. The manuscript is generally well-written, with a comprehensive description of methods in both the main text and is expanded upon in supplementary materials. The motivation behind the study is clear, and the results are both interesting and contextualized effectively within the current limitations of our understanding of LASV transmission in endemic regions.

The authors are transparent about the key assumptions underpinning their modeling and interpretation of results. However, the literature review component of the manuscript could be better described, and its role in identifying key covariates and parameter estimates should be more explicitly justified.

While there are strong assumptions associated with the chosen methods, which may not fully reflect the complexity of LASV transmission, the analysis is robust and provides an excellent foundation for refinement as higher-quality observational data become available. This work will undoubtedly serve as a valuable resource within the scientific literature on Lassa fever. I recommend the manuscript for publication but suggest addressing the following comments to strengthen the study further.

I have attempted to run the code provided in the linked github repository but ran into some difficulties as a file required for analysis has not be provided (the priors for step2_ results/prior_iceberg.RData). The authors have made the underlying data available.

Major comment

The manuscript frequently highlights substantial differences in observed seroprevalence at the village level, even within the same administrative level 2 region. This suggests that aggregating FOI at the chosen level of analysis may overlook critical heterogeneities. While the framing of the study as identifying potential vaccine trial sites at administrative levels 1 or 2 partly justifies this choice, the rationale for selecting this level of aggregation could be made more explicit. Specifically, the manuscript would benefit from a discussion of why a pixel-based approach was not used. Such an approach might better capture intra-regional variability, reduce the impact of administrative unit size on results, and more effectively represent heterogeneity in the included covariates. I am not asking for the analysis to be refactored at this level but I do believe a discussion of consideration of this would be suitable.

Minor comments

Abstract:

Line 25. You estimate the FOI of LASV in human populations driven by transmission from the zoonotic host (M. natalensis, but potentially other rodent and shrew species). Should this more explicitly be termed the force of infection from rodents to humans? Rodent-to-rodent transmission (and therefore a rodent level FOI) occurs in the same settings and is likely important to the human risk of infection but is not modelled in your approach. Similarly the role of human-to-human transmission is likely underestimated in community settings and not explicitly modelled in your approach. There is also increasingly discussion of reverse zoonosis (human-to-rodent FOI), particularly based on studies conducted in Guinea.

Author summary:

Line 43. Replace the oft-cited estimates from reference 3 with the more recently modeled estimates in reference 25. While you critique reference 25 in your discussion, its methodology is more transparent than that of reference 3, which lacks any detail on how its estimates were derived.

Introduction:

Line 60. Again this statement is derived from reference 3. The 20% is based on 48 known infections of which 9 were identified as ill. This is not equivalent to severe, life-threatening hemorrhagic illness and it may be more suitable to report this as “up to 20% of infections are thought to result in symptoms ranging from mild (i.e., headache, fever) through to severe (i.e., coma, acute renal failure) or fatal hemorrhagic illness”. There are several publications from the NCDC where they report disease outcomes (for example, https://pmc.ncbi.nlm.nih.gov/articles/PMC6537738/). While you highlight this further in Line 127-129 it may be worth moving this to this section.

Line 61. Provide a national-level case fatality rate (CFR) from a single year for better clarity. For instance, the NCDC reported a CFR of 17.9% in 2023 based on 1,270 confirmed cases from 28 states (https://ncdc.gov.ng/ncdc.gov.ng/themes/common/files/sitreps/60b4a539bd9b9852ac1a5059ec0f3433.pdf). As the CFR is highly dependent on case detection some states within Nigeria report CFR as 100%.

Line 62-65. See comment above about using updated estimates.

Line 77-78. I assume the description of an ideal study site is derived from CEPI material but it would appear to me that an ideal study site will also need an ongoing assessment of pathogen circulation in rodents to quantify risk of spillover into human populations. As it is unclear how the “presence of frequent LASV spillover” will otherwise be assessed.

Line 86-89. This is potentially true but this is unlikely the sole reason these cases are not detected. Access to healthcare in the locations in which LASV spillovers occur is highly variable across the region, with time varying components (e.g., difficulty in travel due to road conditions, availability to leave farming communities during harvesting periods etc.) or financial constraints (e.g., direct- and indirect costs of healthcare). It would be useful to include some non-clinical considerations in lack of case detection.

Line 113-116. Abundance of rodents and Infection prevalence in rodents is also temporally heterogeneous within small spatial scales (i.e., at village level) not only in a seasonal pattern across broad spatial scales. Further, rodent detection rates =/= rodent abundance which complicates our understanding of rodent ecology and pathogen prevalence within these settings. A detailed description of M. natalensis ecology and our current uncertainties in the effect of this on spillover rates is beyond the scope of this manuscript. However, a sentence around rodent ecology and the rodent-human contact nexus within these contexts would fit well with your mention of “individual and community-level risk factors for infection”.

Methods:

Epidemiological data:

For clarity it may make sense to have subheadings in this section for infection data and case data. This may also make it clearer later when referencing which dataset the reported model outcomes are based on.

Line 161-169. It is unclear what the final epidemiological dataset that was used is. In S1 of reference 61 11 references are included (1 suggests NCDC situation report data with another being ProMED reports (which are no longer available)). You later report excluding all those conducted prior to 1980 (it is not reported why you have chosen this cutoff), which leaves 6 of the 11 references as potentially included. I believe the final dataset is this (https://github.com/mooresea/lassa-model/blob/main/data/case_reports_Lassa_CASES.csv) for cases and (https://github.com/mooresea/lassa-model/blob/main/data/case_reports_Lassa_IGG.csv) for infection, it may be worth directly referencing these in the text as readers may not go to the github repository otherwise. How did you de-duplicate data or decide on which values to use? For example the Redding dataset is derived from NCDC data which you use for previous years and in the past I have had difficulties reconciling case counts from Liberia across multiple publications.

Clarify the reasoning for excluding studies prior to 1980 and whether this cutoff impacts representativeness.

Covariate data:

Did you assess collinearity among covariates to mitigate potential confounding effects? This may be particularly relevant for the travel-time-to-healthcare and land-use covariates. The statistical models are described in more depth in the supplementary material, it appears that a large number of potential covariates for limited observational data were used.

Model:

Line 233-235. Is it reasonable to assume a constant FOI over time given fluctuations in rodent abundance and LASV prevalence within the rodent populations? I understand that a time-varying FOI may not be suitable given data resolution limitations and computational burdens but I think if this is a necessary simplifying assumption this should be expanded upon further in the methods section or discussion.

Line 278-287. If I understand this correctly this section is about inferring the number of infections that would be symptomatic. Is it worth discussing that the ENABLE study presumably has intensive investigation of symptoms for individuals found to seroconvert/reporting of symptoms and subsequent testing and may not represent what might be considered a symptomatic disease in the absence of an intensive prospective cohort study, potentially overestimating the rate of symptomatic disease?

Results:

Line 328-341. You introduce an in-depth review of the published literature without previously describing it in the methods. Is this section in reference to Lines 167-172 in the epidemiological data section? If so I think there needs to be a bit more information on how these were selected? Was this based on a systematic search and review of the literature or a review of author identified research? How did you assess whether this was a comprehensive set of manuscripts? It is unclear how studies of imported LF cases would be helpful in inferring FOI in the endemic region.

Line 336-338. Studies from Guinea have fairly consistently shown higher prevalence of infection among rodent populations and consistent evidence of LASV prevalence. However, this may not be the case across the endemic region, it may be worth highlighting examples of the inverse, where LASV may not be circulating consistently (i.e., https://pmc.ncbi.nlm.nih.gov/articles/PMC7824740/).

Line 367-370. Clarify whether the "average" reported is the mean or median. A median may be more appropriate, given the likely non-normal distribution of probabilities.

Line 396-7. Incomplete sentence? “. center,”

LASV infection attack rates and incidence and figure 6. I am a bit concerned about the age stratified results. Based on published data seroprevalence generally increases with age but the rate of increase varies with site. Despite this the proportion of cases that are detected in individuals under 18 is a minority of reported cases. To my knowledge there is not much information about the varying symptomatology by age but stratifying your results into <12, 12-17 and 18+ suggests that we’d be expecting most symptomatic disease to occur in <12 year olds. Can I suggest showing your results in a different way? Perhaps a figure of LASV incidence (infection) or LF incidence (disease) with age on the x-axis (continuous time or 5-10 year age bands) and annual incidence per 1,000 on the y-axis. The different sites can be represented by colours with associated credible/confidence intervals. Or perhaps panelling by sites if it becomes difficult to interpret overlain colours. In this way I think it would be easier to appreciate that the rate of infection or disease varies by time but there is a consistent pattern across some sites.

Seasonality. The associated figure (fig 8.) could be moved to supplementary. It doesn't appear that this is a new analysis and contains a regression model fitted to reported case counts which have previously been discussed to represent substantial underreporting. The figure should be titled seasonality of LF case reporting rather than incidence. I don’t think much of the data is based on the date of symptom development of the identified cases and is instead the epi-week or month in which they have been added to the relevant line list. There is a wide range of incubation period and symptom onset reported in the literature (https://doi.org/10.1016/S2214-109X(24)00379-6).

Discussion:

It may be worth including a line in the discussion around line 583-586 about the length of time vaccine trials may need to be completed over (i.e., years) to account for the seasonal dynamics expected in spillover events/human cases.

Not too sure if this is more relevant for the methods section or limitations within the discussion but case definitions over the study period will vary by country, time period and diagnostic test availability. Thus there may be a time varying surveillance bias/undercounting of cases.

Line 556: “Locations with high baseline seroprevalence may experience few LF cases even if LASV is actively circulating in the rodent population.” This is an important point and while the subsequent impact of vaccinating a population in these contexts is beyond the scope of this manuscript it highlights that rodent pathogen dynamics should be included in the introduction as it is an important component of the disease system.

Line 610-620: Current hotspots are commonly associated with specialist treatment centers, awareness of the disease, surveillance systems and available diagnostics. Much of the remaining endemic area does not have access to these and so it may remain premature to rule out a similar burden (~50 confirmed annual deaths at adm1 level, presumably single digit reported deaths at adm2) in non-traditional hotspots.

Code review:

I ran the code for admin level 2 and a reversion rate of 6.

Update readme or script step1 to explain requirement to specify the level of analysis admin1/2 and seroreversion rate.

This code produced an error for me.

for(ii in 1:nrow(s)){

a[ii,] = a.tmp[which(paste(a.tmp$ADM1_code,a.tmp$YEAR) ==

paste(s$ADM1_code,pmin(2014,s$YEAR))[ii]),] }

Error in x[[jj]][iseq] <- vjj : replacement has length zero

I achieved the intended outcome with a <- merge(s, a.tmp, by.x = c("ADM1_code", "YEAR_adj"), by.y = c("ADM1_code", "YEAR"), all.x = TRUE)

Update readme or comment line 20 in step2 to direct the user to extract data/adm_2_pop_upd.csv

step2_estimate_… line 72 priors have not been uploaded to the repo and it is unclear the structure they take so I have been unable to run the remaining script as these are necessary parameters for the subsequent functions.

step3_ unable to run as required results from step_2. The code looks like it performs the expected analysis.

Unable to run step4_ beyond line 46

Haven’t run the remaining step4_/step5_ or step6_ scripts and challenging to check them based on code as written.

Overall, this is an important and timely manuscript that makes a valuable contribution to the field. Thank you for the opportunity to review it.

**Figure resubmission:****Reproducibility:** To enhance the reproducibility of your results, we recommend that authors of applicable studies deposit laboratory protocols in protocols.io, where a protocol can be assigned its own identifier (DOI) such that it can be cited independently in the future. Additionally, PLOS ONE offers an option to publish peer-reviewed clinical study protocols. Read more information on sharing protocols at https://plos.org/protocols?utm_medium=editorial-email&utm_source=authorletters&utm_campaign=protocols

---

## [Decision Letter · Decision Letter 1]

Dear Dr. Moore,

We are pleased to inform you that your manuscript 'Estimation of Lassa fever incidence rates in West Africa: development of a modeling framework to inform vaccine trial design' has been provisionally accepted for publication in PLOS Neglected Tropical Diseases.

Best regards,

Ran Wang, M.D.

Academic Editor

Mabel Carabali

Section Editor

Shaden Kamhawi

co-Editor-in-Chief

Paul Brindley

co-Editor-in-Chief

Reviewer's Responses to Questions

**Key Review Criteria Required for Acceptance?**

**Methods**

-Are the objectives of the study clearly articulated with a clear testable hypothesis stated?

-Is the study design appropriate to address the stated objectives?

-Is the population clearly described and appropriate for the hypothesis being tested?

-Is the sample size sufficient to ensure adequate power to address the hypothesis being tested?

-Were correct statistical analysis used to support conclusions?

-Are there concerns about ethical or regulatory requirements being met?

Reviewer #3: Improved description of methods from previous

**Results**

-Does the analysis presented match the analysis plan?

-Are the results clearly and completely presented?

-Are the figures (Tables, Images) of sufficient quality for clarity?

Reviewer #3: Results are now more clearly presented.

**Conclusions**

-Are the conclusions supported by the data presented?

-Are the limitations of analysis clearly described?

-Do the authors discuss how these data can be helpful to advance our understanding of the topic under study?

-Is public health relevance addressed?

Reviewer #3: Conclusions are justified.

**Editorial and Data Presentation Modifications?**

Reviewer #3: NA

**Summary and General Comments**

Reviewer #3: The authors have satisfactorily responded to my previous comments. I have no concerns about this being accepted in its current condition.

I have a minor comment about the use of ring-vaccination the authors may want to consider but it is up to them. For ring vaccination to be a suitable method to guide vaccine efficacy trials the incubation period and delay in health seeking and subsequent confirmatory testing would need to be overcome. For example, delay from spillover infection to implementation of ring vaccination of 4 weeks (1-7 days incubation time, 7-14 days symptom progression and presentation for testing, 1-4 days confirmatory testing and reporting, 1-4 implementation of protocol at study site) may result in the trial arriving at the location following an outbreak but when there is no further active spillover. It could be a requirement of ring vaccination that active surveillance, sensitisation and improved reporting are in place prior to adopting this approach in suitable administrative regions.

I look forward to seeing this manuscript guide future discussion around vaccine trial design for emerging zoonoses.

PLOS authors have the option to publish the peer review history of their article (what does this mean? ). If published, this will include your full peer review and any attached files.

**Do you want your identity to be public for this peer review?** For information about this choice, including consent withdrawal, please see our Privacy Policy .

Reviewer #3: **Yes: ** David Simons

---

## [Editor Report · Acceptance letter]

Dear Dr. Moore,

We are delighted to inform you that your manuscript, "Estimation of Lassa fever incidence rates in West Africa: development of a modeling framework to inform vaccine trial design," has been formally accepted for publication in PLOS Neglected Tropical Diseases.

Best regards,

Shaden Kamhawi

co-Editor-in-Chief

Paul Brindley

co-Editor-in-Chief
